# Green leaf volatile sensory calcium transduction in *Arabidopsis*

Yuri Aratani[1,5], Takuya Uemura[1,5], Takuma Hagihara [1], Kenji Matsui [2] & Masatsugu Toyota [1,3,4] ✉

Plants perceive volatile organic compounds (VOCs) released by mechanically- or herbivore-damaged neighboring plants and induce various defense responses. Such interplant communication protects plants from environmental threats. However, the spatiotemporal dynamics of VOC sensory transduction in plants remain largely unknown. Using a wide-field real-time imaging method, we visualize an increase in cytosolic $Ca^{2+}$ concentration ($[Ca^{2+}]_{cyt}$) in *Arabidopsis* leaves following exposure to VOCs emitted by injured plants. We identify two green leaf volatiles (GLVs), ($Z$)-3-hexenal ($Z$-3-HAL) and ($E$)-2-hexenal ($E$-2-HAL), which increase $[Ca^{2+}]_{cyt}$ in *Arabidopsis*. These volatiles trigger the expression of biotic and abiotic stress-responsive genes in a $Ca^{2+}$-dependent manner. Tissue-specific high-resolution $Ca^{2+}$ imaging and stomatal mutant analysis reveal that $[Ca^{2+}]_{cyt}$ increases instantly in guard cells and subsequently in mesophyll cells upon $Z$-3-HAL exposure. These results suggest that GLVs in the atmosphere are rapidly taken up by the inner tissues via stomata, leading to $[Ca^{2+}]_{cyt}$ increases and subsequent defense responses in *Arabidopsis* leaves.

Plants emit an array of volatile organic compounds (VOCs), including green leaf volatiles (GLVs), terpenoids, and amino acid derivatives, in response to wounding and herbivore attack[1,2]. These VOCs exert multiple protective effects, such as directly repelling herbivores and attracting natural enemies of the herbivores[3–5]. Neighboring intact plants perceive these VOCs as danger cues to trigger defense responses[6] or prime themselves to respond to upcoming stresses in a timely manner[7]. Therefore, these VOCs serve as interplant airborne signals. Such interplant interaction mediated by volatile cues is known as plant–plant communication or plant eavesdropping[1]. VOC perception in plants was first reported in the early 1980s. Two individual groups demonstrated that Sitka willow (*Salix sitchensis*) and poplar (*Populus x euramericana*) trees exhibited increased antiherbivore properties when grown near damaged plants[8,9]. Such VOC-mediated interplant signaling has been reported in more than 30 plant species, including lima bean[6], tobacco[10], tomato[11], sagebrush[12], and *Arabidopsis*[13].

GLVs, comprising six-carbon (C6) compounds, including alcohols, aldehydes, and esters, are among the most rapidly and abundantly produced VOCs in the plant kingdom, being generated within seconds after wounding[14]. Upon mechanical damage, GLV biosynthesis is immediately initiated by a lipoxygenase (LOX)-mediated dioxygenase reaction on fatty acids to yield fatty acid/lipid hydroperoxides. These derivatives are cleaved to form ($Z$)-3-hexenal ($Z$-3-HAL), a C6 volatile aldehyde, by hydroperoxide lyase (HPL), a key enzyme for GLV formation[14]. *Arabidopsis* accession Col-0 exhibits low GLV production because of a 10-bp deletion in *HPL*[15]. $Z$-3-HAL can be isomerized to ($E$)-2-hexenal ($E$-2-HAL) by an isomerase[16,17]. Alternatively, $Z$-3-HAL is quickly reduced to the C6 alcohol ($Z$)-3-hexenol ($Z$-3-HOL), which can be further converted to ($Z$)-3-hexenyl acetate ($Z$-3-HAC) by an acetyl transferase. GLVs elicit a wide range of defense signals in plants. GLV exposure in plants induces the accumulation of the stress-related phytohormone jasmonic acid (JA)[7] and expression of JA-dependent

[1]Department of Biochemistry and Molecular Biology, Saitama University, Saitama 338-8570, Japan. [2]Graduate School of Sciences and Technology for Innovation, Yamaguchi University, Yamaguchi 753-8515, Japan. [3]Suntory Rising Stars Encouragement Program in Life Sciences (SunRiSE), Suntory Foundation for Life Sciences, Kyoto 619-0284, Japan. [4]Department of Botany, University of Wisconsin, Madison, WI 53706, USA. [5]These authors contributed equally: Yuri Aratani, Takuya Uemura. ✉e-mail: mtoyota@mail.saitama-u.ac.jp

defense genes[18] or plants prime themselves for rapid response to subsequent herbivory damage[19]. GLVs also trigger the induction of abiotic stress-related defense genes, including heat- and oxidative stress-responsive genes[20]. Indeed, treating *Arabidopsis* with GLVs induces enhanced heat stress tolerance[21].

Plant VOC-dependent defense responses are initiated following VOC entry into inner tissues[22]. Exogenous VOC uptake by plants is facilitated during the light period, in which stomata are opened and $CO_2$ exchange is enhanced[23]. Furthermore, changes in the uptake rate of aldehydes and ketones can be coupled with those of stomatal conductance[24], suggesting that stomata play a critical role in VOC perception. In tobacco, TOPLESS-like protein, a transcriptional corepressor for JA signaling, was revealed to interact with the sesquiterpene β-caryophyllene[25]. Upon binding to β-caryophyllene, the TOPLESS-like protein-mediated transcriptional suppression of defense-related genes is inhibited, thereby accelerating defense gene expression[25]. However, the molecular mechanisms underlying the perception of VOCs, especially GLVs, are poorly understood.

Cytosolic $Ca^{2+}$ plays crucial roles in a wide array of plant stress responses[26,27]. Downstream signaling activation systems via cytosolic calcium ion concentration ($[Ca^{2+}]_{cyt}$) increases have been extensively studied, including the identification of critical components linking stress perception to the formation of $Ca^{2+}$ signals, as well as the mechanism of $Ca^{2+}$ signal implementation[28]. Using an electrophysiological technique and a $Ca^{2+}$-sensitive fluorescent dye, membrane potential depolarization and $[Ca^{2+}]_{cyt}$ increases after exposure to several GLVs were observed in tomato leaves[29]. In addition, $[Ca^{2+}]_{cyt}$ increases were detected in detached *Arabidopsis* leaves upon VOC exposure using a $Ca^{2+}$-sensitive luminescent protein[30]. These findings suggest that $Ca^{2+}$ signaling is involved in an early process leading to downstream defense responses. However, little is known about the spatiotemporal dynamics of GLV sensory transduction because of technical limitations in real-time monitoring of GLV-induced $Ca^{2+}$ signals in intact living plants.

In this study, we observed that a plant-wide $Ca^{2+}$ signature rapidly occurs in response to exposure to VOCs released by damaged plants using transgenic *Arabidopsis* expressing a green fluorescent protein-based $Ca^{2+}$ biosensor and a wide-field real-time fluorescence microscope. We found that *Z*-3-HAL and *E*-2-HAL elicited rapid increases in $[Ca^{2+}]_{cyt}$ in *Arabidopsis* leaves, in which defense responses were activated. Tissue-specific high-resolution $Ca^{2+}$ imaging and stomatal mutant analysis also clarified the spatiotemporal dynamics of VOC sensing and signal transduction networks in plants.

## Results

### VOCs released from plants trigger $[Ca^{2+}]_{cyt}$ increases in *Arabidopsis*

Using transgenic *Arabidopsis* expressing the $Ca^{2+}$ biosensor GCaMP3[31], we observed $[Ca^{2+}]_{cyt}$ changes in intact plants following exposure to VOCs emitted by damaged plants in real time. We used *Arabidopsis* accession No-0 and tomato as emitter plants because they emit various VOCs, including GLVs, in response to wounding[29,32,33]. First, we monitored $[Ca^{2+}]_{cyt}$ increases in *Arabidopsis* (receiver) following exposure to VOCs emitted from *Arabidopsis* plants (source of VOCs) fed on by the common cutworm (*Spodoptera litura*; Fig. 1a, b and Supplementary Movie 1). This $Ca^{2+}$ signal was rapidly transmitted to all parts of the plant within 20 min (Fig. 1b, c). Similar results were obtained when receiver *Arabidopsis* was exposed to VOCs released by tomato leaves (source of VOCs) consumed by *S. litura* (Fig. 1b, c and Supplementary Movie 2). Further, upon placing either homogenized *Arabidopsis* or tomato leaves (Source of VOCs) near receiver *Arabidopsis* (Fig. 2a), $[Ca^{2+}]_{cyt}$ increases were observed in several leaves (Fig. 2b, Supplementary Movies 3 and 4). In a time-course analysis of $[Ca^{2+}]_{cyt}$ changes in leaf 1 (L1), we found that VOCs emitted by homogenized tomato leaves caused larger signal changes than those emitted by *Arabidopsis*

(Fig. 2c, d). These results suggest that VOCs emitted by damaged *Arabidopsis* and tomato plants caused $[Ca^{2+}]_{cyt}$ changes in neighboring intact *Arabidopsis* plants.

### *Z*-3-HAL and *E*-2-HAL induce $[Ca^{2+}]_{cyt}$ increases in *Arabidopsis*

To identify the VOCs responsible for triggering $[Ca^{2+}]_{cyt}$ changes in *Arabidopsis*, we examined five GLVs [*Z*-3-HAL, *E*-2-HAL, *n*-hexanal (*n*-HAL), *Z*-3-HOL, and *Z*-3-HAC], three terpenes (α-pinene, β-pinene, and β-caryophyllene), and methyl jasmonate (MeJA) because they have been reported to induce defense responses in receiver plants[2,11,29]. We measured time-course changes in $[Ca^{2+}]_{cyt}$ in three regions, i.e., "tip," "base," and the midpoint between them ("mid"), in L1 of *Arabidopsis* after each VOC solution was placed in close proximity (Fig. 3a). Of these, *Z*-3-HAL immediately increased $[Ca^{2+}]_{cyt}$ at the tip within 30 s (Fig. 3b, c and Supplementary Movie 5). Subsequently, the signal was propagated to the mid and base regions within 1 and 2 min, respectively, and it persisted for over 15 min. Although exposing *Arabidopsis* to *E*-2-HAL also caused a $[Ca^{2+}]_{cyt}$ increase, the increase was weaker and slower than that induced by *Z*-3-HAL (Fig. 3b–d and Supplementary Movie 6). By measuring the velocities of *Z*-3-HAL- and *E*-2-HAL-induced $Ca^{2+}$ transmission, we revealed that the signal transmission induced by *Z*-3-HAL (0.24–0.30 mm/s; $N = 10$) was faster than that induced by *E*-2-HAL (0.01–0.02 mm/s; $N = 10$; Fig. 3e). In contrast, no $[Ca^{2+}]_{cyt}$ increases were observed in leaves exposed to other VOCs (Fig. 3f). These results indicate that *Z*-3-HAL and *E*-2-HAL are key triggers of rapid $[Ca^{2+}]_{cyt}$ increases in *Arabidopsis*. Based on these observations, we focused on *Z*-3-HAL and *E*-2-HAL in subsequent analyses.

To confirm that GLVs induce $[Ca^{2+}]_{cyt}$ increases upon exposure to VOCs emitted by homogenized leaves (Fig. 2), we monitored $[Ca^{2+}]_{cyt}$ changes in leaves following exposure to VOCs emitted by homogenized *Arabidopsis* that harbors inactive *HPL* and emits little GLVs[34]. VOCs emitted by homogenized *Arabidopsis* carrying Col-0-derived *HPL* (*hpl1* mutant) did not induce $Ca^{2+}$ signals in receiver leaves (Supplementary Fig. 1). Using gas chromatography-mass spectrometry (GC-MS) and MonoTrap RGPS TD, which is a high-quality adsorbent, we further analyzed the VOC components rapidly produced by homogenized tomato and *Arabidopsis* leaves. Tomato leaves emitted significantly higher amounts of *Z*-3-HAL and *E*-2-HAL than wild-type (WT) *Arabidopsis* leaves upon homogenization (Supplementary Fig. 2). Moreover, the levels of these compounds emitted by homogenized *hpl1* leaves were remarkably low (<25% of that emitted by WT leaves; Supplementary Fig. 2), consistent with the results of $Ca^{2+}$ signature levels. Altogether, these results suggest that interplant airborne signaling that induces $[Ca^{2+}]_{cyt}$ changes is dependent on *HPL*-mediated GLV formation in *Arabidopsis*.

### *Z*-3-HAL and *E*-2-HAL elicit electrical signals and defense gene expression

Because changes in the membrane potential upon GLV exposure were observed in tomato leaves[29], we simultaneously recorded the changes in $[Ca^{2+}]_{cyt}$ and the leaf surface potential upon C6 aldehyde exposure in *Arabidopsis*. *Z*-3-HAL and *E*-2-HAL exposure in *Arabidopsis* leaves resulted in rapid changes in the leaf surface potential, which is spatiotemporally coupled with changes in $[Ca^{2+}]_{cyt}$ (Fig. 4a–d). Interestingly, detailed analysis of the timing of the initial detectable signal changes revealed that the surface potential change significantly preceded the onset of the $Ca^{2+}$ signal in the plant GLV sensory transduction system (Fig. 4e).

We next examined the accumulation of defense-related gene transcripts after *Z*-3-HAL and *E*-2-HAL exposure. Expression of the heat and oxidative stress response marker genes *HSP90.1* and *ZAT12* increased in leaves after 30 and 60 min of *Z*-3-HAL or *E*-2-HAL exposure (Fig. 4f). Similarly, JA-related genes, such as *OPR3* and *JAZ7*, were upregulated by these C6 aldehydes (Fig. 4f). Furthermore, we used a pharmacological approach to assess the role of GLV-induced $[Ca^{2+}]_{cyt}$

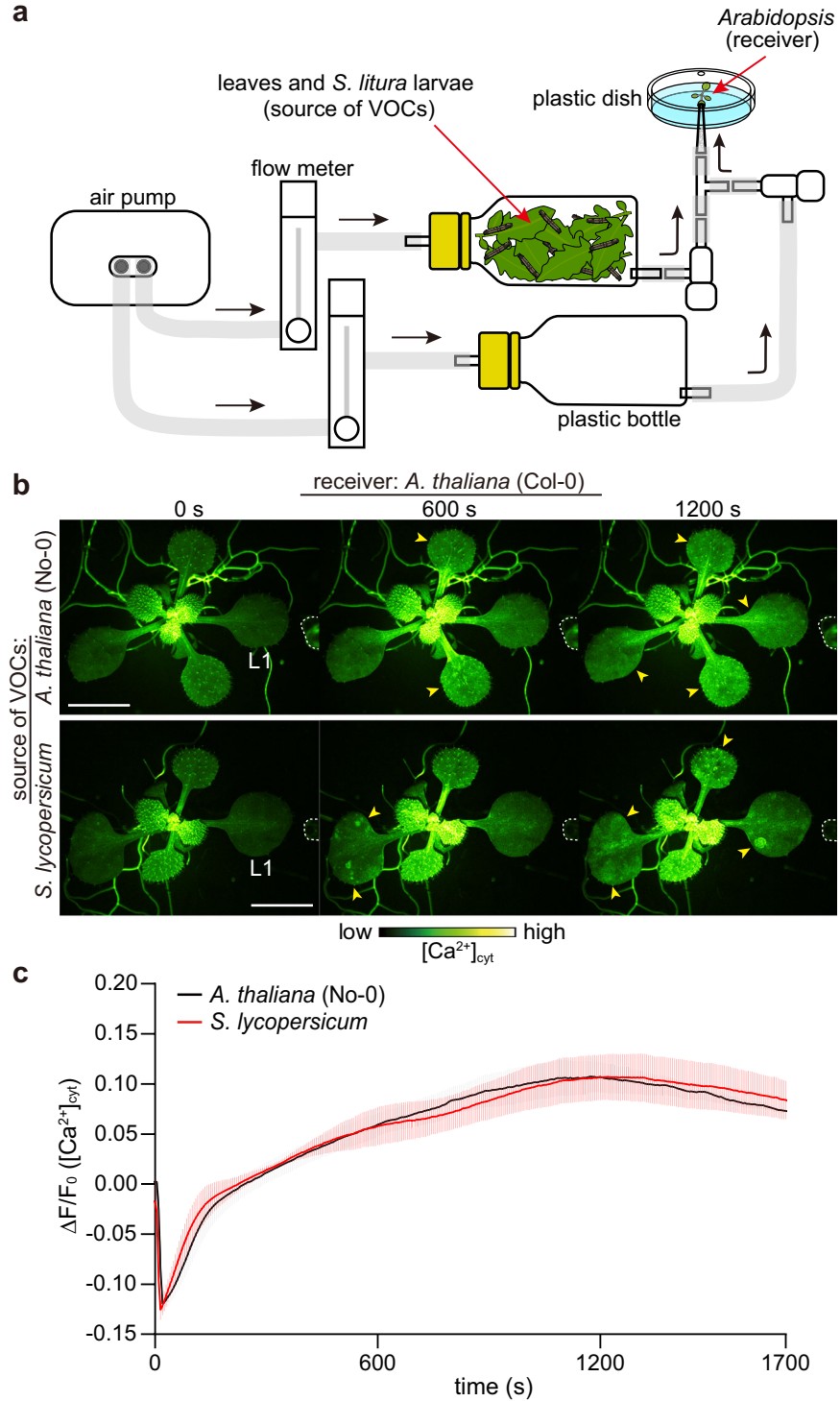

**Fig. 1 | Exposure to VOCs emitted by leaves consumed by the herbivore *S. litura* induces [Ca²⁺]cyt increases in receiver *Arabidopsis* leaves. a** The experimental setup for Ca²⁺ imaging in *Arabidopsis* (receiver) upon exposure to VOCs emitted by leaves consumed by *S. litura* larvae is schematically illustrated. Prior to the experiment, the receiver *Arabidopsis* in a plastic dish was acclimated by directing airflow from an empty plastic bottle for 10 min, allowing its adaptation to the experimental conditions. Subsequently, receiver *Arabidopsis* was exposed to VOCs emitted from a plastic bottle containing *S. litura* larvae and either *Arabidopsis* or tomato leaves (source of VOCs) by connecting the bottle and manipulating the valve. The black arrows indicate the direction of airflow. **b** Changes in [Ca²⁺]cyt (yellow arrowheads) in *Arabidopsis* expressing GCaMP3 in response to VOCs released from *Arabidopsis* (upper) and tomato (*Solanum lycopersicum* cv. Minicarol) leaves (below) consumed by *S. litura* larvae. White dashed lines indicate the position of the tip of the tube from where the airflow emerges. Scale bar, 5 mm. **c** Quantification of [Ca²⁺]cyt signatures in leaf 1 (L1). Error bars, mean ± standard error (SE). *N* = 4 and 5 biologically independent samples for *A. thaliana* and *S. lycopersicum*, respectively.

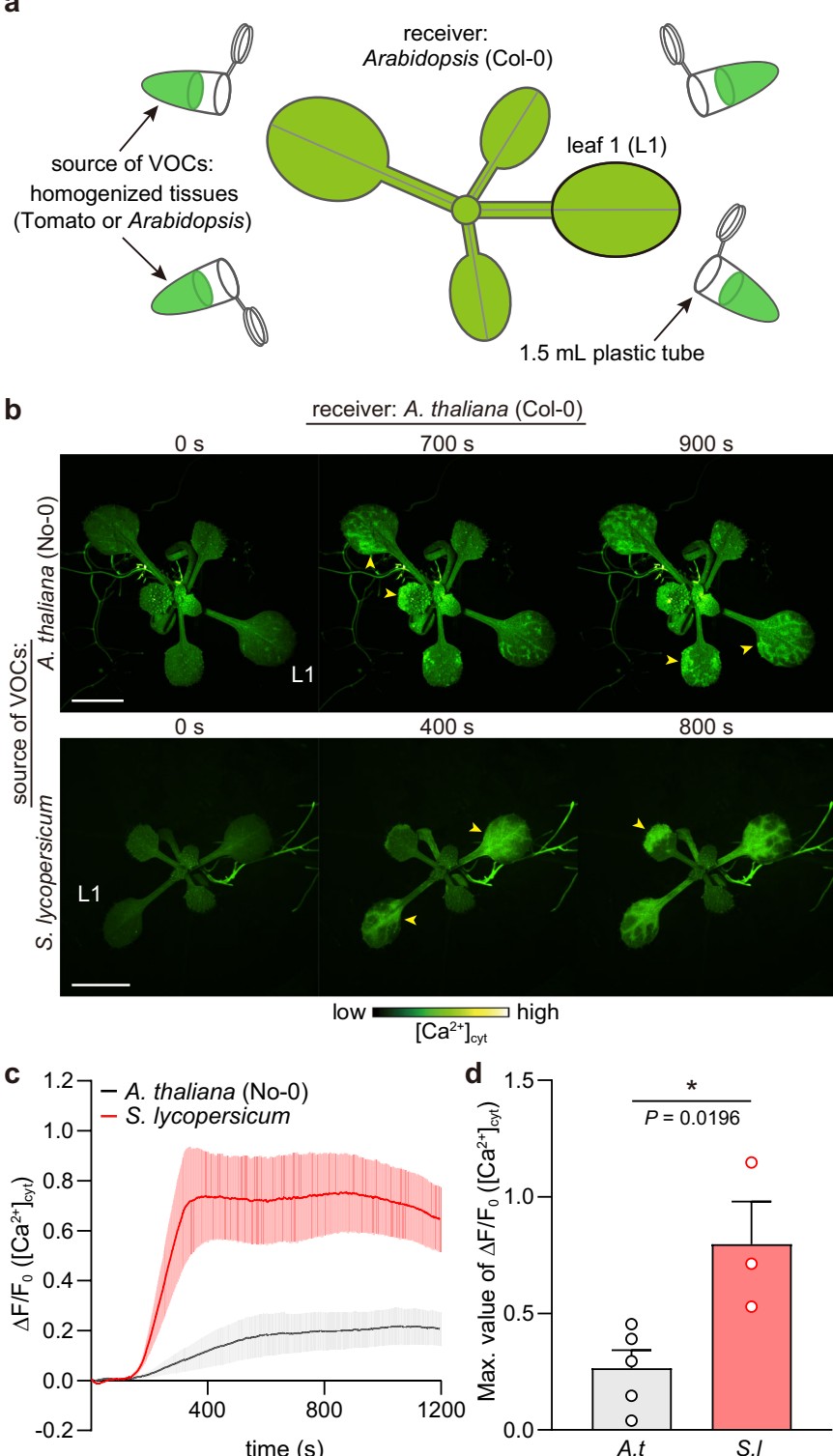

**Fig. 2 | Exposure to VOCs emitted by homogenized leaves induces [Ca²⁺]$_{cyt}$ increases in receiver *Arabidopsis* leaves. a** The experimental setup for Ca²⁺ imaging in *Arabidopsis* (receiver) upon exposure to VOCs from homogenized *Arabidopsis* or tomato leaves (source of VOCs) is schematically illustrated. In total, 10 g of *Arabidopsis* leaves or 5 g of tomato leaves was homogenized with liquid nitrogen using a mortar and pestle. The resulting homogenized tissues were immediately transferred to 1.5-mL plastic tubes. Subsequently, Ca²⁺ imaging was initiated by placing the tubes in close proximity to receiver *Arabidopsis*. **b** Changes in [Ca²⁺]$_{cyt}$ (yellow arrowheads) in *Arabidopsis* expressing GCaMP3 after exposure to VOCs emitted by homogenized *Arabidopsis* (upper) and tomato (*Solanum lycopersicum* cv. Micro-Tom) leaves (lower). Scale bar, 5 mm. **c** Quantification of [Ca²⁺]$_{cyt}$ signatures in L1. Error bars, mean ± SE. *N* = 5 and 3 biologically independent samples for *A. thaliana* and *S. lycopersicum*, respectively. **d** Comparison of the maximal [Ca²⁺]$_{cyt}$ changes detected in receiver *Arabidopsis* upon exposure to VOCs emitted by homogenized *Arabidopsis* (*A.t*) or tomato (*S.l*) leaves. An asterisk denotes statistically significant differences based on two-tailed Student's *t*-test (*, *P* < 0.05). Error bars, mean ± SE. *N* = 5 and 3 biologically independent samples for *A. thaliana* and *S. lycopersicum*, respectively.

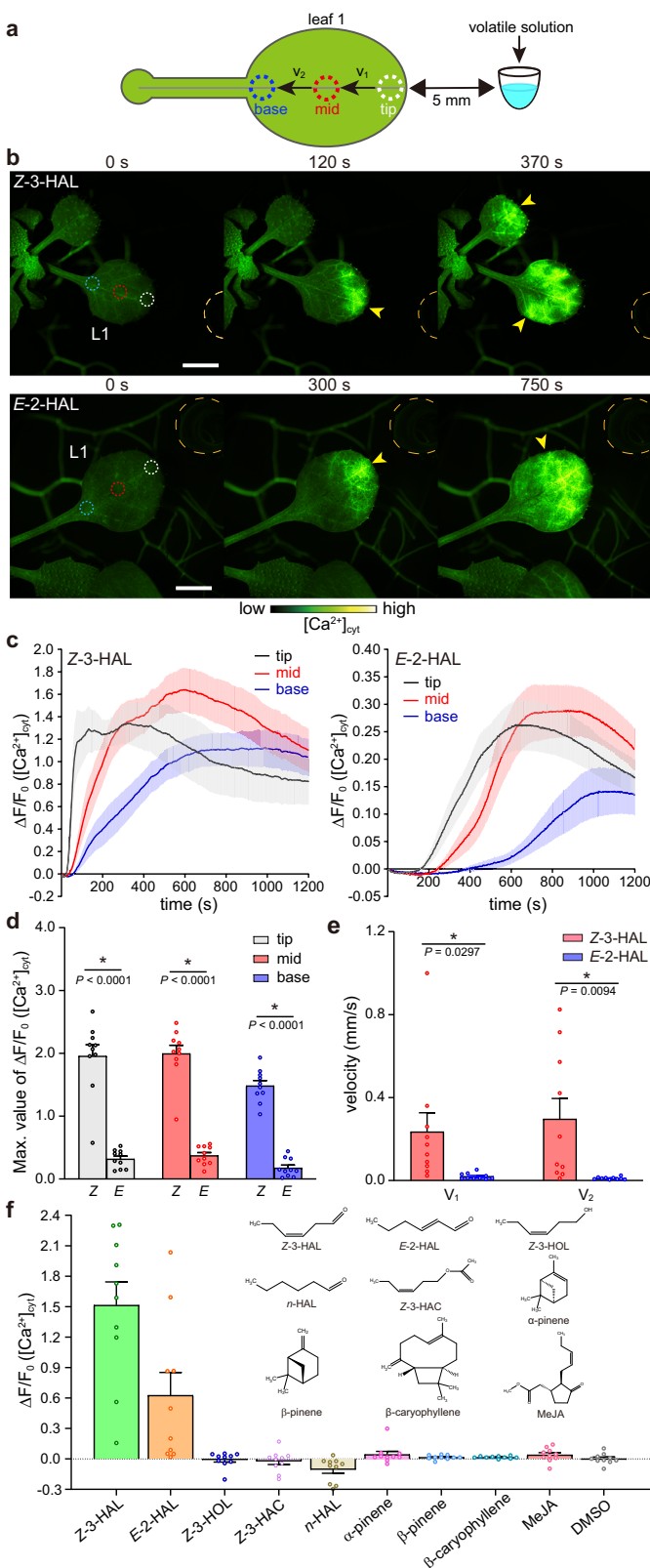

**Fig. 3 | Z-3-HAL and E-2-HAL trigger [Ca²⁺]$_{cyt}$ increases in *Arabidopsis*. a** Diagram presenting an *Arabidopsis* leaf and the regions of interest (ROIs) used to analyze [Ca²⁺]$_{cyt}$ changes and velocities (V$_1$ and V$_2$). Ten microliters of each chemical solution were applied to a plastic tube placed 5 mm from the tip region of *Arabidopsis* leaf 1 (L1). [Ca²⁺]$_{cyt}$ changes at each point (tip, mid, and base) were analyzed over time. **b** Time-course changes in [Ca²⁺]$_{cyt}$ (yellow arrowheads) in the L1 of *Arabidopsis* after applying *Z*-3-HAL (upper) and *E*-2-HAL (lower) at a distance of 5 mm from the tip of L1. The chemical solution was applied into a plastic tube, indicated by an orange dashed line (0 s). Dashed white, red, and blue circles indicate the position of the tip, mid, and base regions, respectively. Scale bar, 2.5 mm. **c** Quantification of the [Ca²⁺]$_{cyt}$ signature in the tip, mid, and base regions induced by *Z*-3-HAL (left) and *E*-2-HAL (right). Error bars, mean ± SE. $N = 10$ biologically independent samples. **d** Comparison of the maximal [Ca²⁺]$_{cyt}$ changes detected in each ROI upon exposure to *Z*-3-HAL and *E*-2-HAL. Error bars, mean ± SE. $N = 10$ biologically independent samples. An asterisk denotes statistically significant differences based on two-tailed Student's $t$ test (*, $P < 0.05$). **e** Velocities (mm/s) of the Ca²⁺ signals transmitted between the tip and mid regions (V$_1$) and between the mid and base regions (V$_2$) induced by *Z*-3-HAL and *E*-2-HAL were analyzed. Error bars, mean ± SE. $N = 10$ biologically independent samples. An asterisk denotes statistically significant differences based on two-tailed Student's $t$ test (*, $P < 0.05$). **f** Screening of VOCs that elicit [Ca²⁺]$_{cyt}$ increases in *Arabidopsis*. [Ca²⁺]$_{cyt}$ signatures in the tip region of L1 after 400 s of volatile solution treatment were analyzed. Error bars, mean ± SE. $N = 10$ biologically independent samples.

period of 24 h with a liquid medium lacking LaCl$_3$ and EGTA, we observed approximately 65–95% recovery of the Ca²⁺ signal at 500 s compared with that in Mock-pretreated *Arabidopsis* (Supplementary Fig. 3a, c). Therefore, the GLV response in plants pretreated with the reagents is reversible, pharmacologically suggesting that these chemicals can block Ca²⁺ signals without drastically affecting the plant cells themselves. Altogether, these results indicate that [Ca²⁺]$_{cyt}$ increases are required for the induction of transcriptional changes related to defense responses in plants.

### Z-3-HAL induces local Ca²⁺ signals in a concentration-dependent manner

To gain further insights into the physiological properties of C6 aldehydes as signaling molecules, we examined the concentration dependency of [Ca²⁺]$_{cyt}$ increases induced by *Z*-3-HAL. Upon exposure to lower *Z*-3-HAL concentrations, concentration-dependent decreases of [Ca²⁺]$_{cyt}$ responses were observed (Fig. 5a, b, and Supplementary Fig. 4). We quantified the amount of *Z*-3-HAL released from *Z*-3-HAL solution under this experimental condition. *Z*-3-HAL was adsorbed by adsorbents placed at a distance of 5 mm from the 0.03 M solution. *Z*-3-HAL adsorption reached saturation within 1 min (Supplementary Fig. 5), and 3.85 nmol *Z*-3-HAL was adsorbed over 30 s. Considering the volume of the adsorbents (58 μL), it can be estimated that the local concentration of *Z*-3-HAL at a distance of 5 mm from the 0.03 M solution was 0.07 mM. Based on this finding, L1 in receiver *Arabidopsis* was exposed to approximately 6.42–385 nmol (0.1–6.7 mM) *Z*-3-HAL over 30 s following exposure to 0.05–3.0 M *Z*-3-HAL solutions (Fig. 5a).

As Ca²⁺ has been proposed to act as a long-distance signal traveling to systemic undamaged parts to activate defense responses at a whole-plant level in other stress responses[31,35], we investigated whether *Z*-3-HAL triggers long-distance intracellular Ca²⁺ signal propagation. L1 was spatially isolated from other parts of the plant (Fig. 5c), ensuring that only L1 was exposed to *Z*-3-HAL. Exposure of L1 to *Z*-3-HAL caused a local [Ca²⁺]$_{cyt}$ increase; however, Ca²⁺ signal propagation to distal unstimulated parts (L3) was not observed (Fig. 5d–f and Supplementary Movie 7). These results indicate that *Z*-3-HAL elicits [Ca²⁺]$_{cyt}$ increases locally following direct exposure but not long-distance Ca²⁺ signals traveling toward systemic leaves.

### GLVs are rapidly perceived by guard and mesophyll cells

We generated *Arabidopsis* expressing GCaMP3 driven by tissue-specific promoters, such as *GC1*, *RBCS1A*, *SULTR2;2*, and *ATML1*, to selectively

increases in triggering transcript accumulation. Pretreating *Arabidopsis* seedlings with a Ca²⁺ channel blocker (LaCl$_3$) or calcium chelator (EGTA) prevented both [Ca²⁺]$_{cyt}$ increases and marker gene expression (Supplementary Fig. 3). Furthermore, we conducted the washout assay to evaluate the reversibility of the Ca²⁺ signals in *Arabidopsis*, allowing the assessment of any potential negative effects of these pharmacological reagents on plant cell viability. After an additional incubation

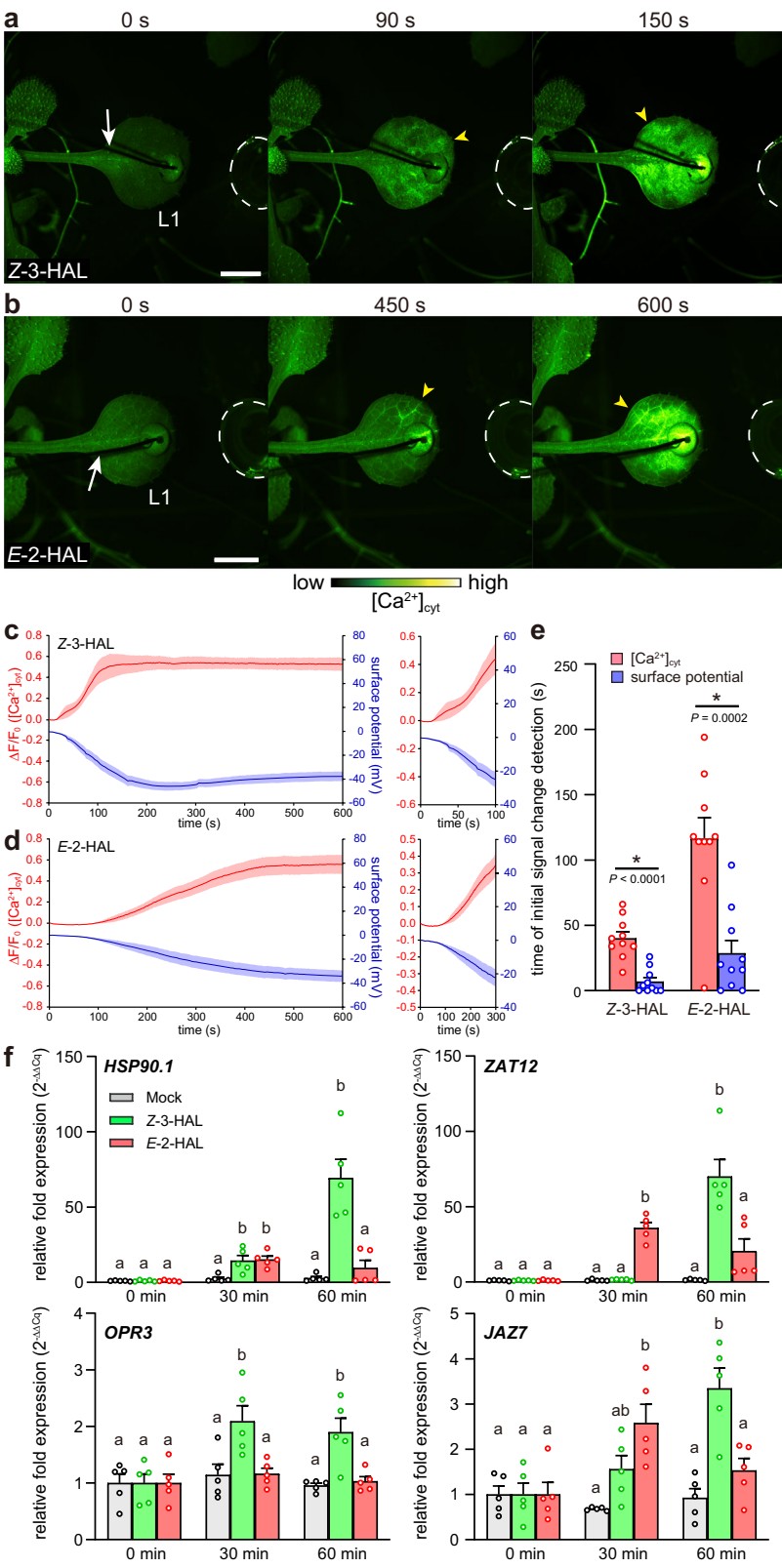

express GCaMP3 in guard[36], mesophyll[37], vasculature[38], and epidermal cells[39], respectively. These transgenic lines enable the visualization of $[Ca^{2+}]_{cyt}$ changes specifically occurring in each tissue, thereby aiding in the understanding of the spatiotemporal patterns of GLV-induced responses. Upon $Z$-3-HAL exposure, $[Ca^{2+}]_{cyt}$ increased rapidly in the tip region of $pGC1::GCaMP3$ and $pRBCS1A::GCaMP3$ leaves within 40 s, whereas it increased gradually in $pSULTR2;2::GCaMP3$ and

$pATML1::GCaMP3$ leaves (Fig. 6 and Supplementary Movies 8–11). Quantitative analysis of the timing of signal increases revealed that $[Ca^{2+}]_{cyt}$ first increased in $pGC1::GCaMP3$ and $pRBCS1A::GCaMP3$ within 1 min (Fig. 6f, g). Interestingly, $pSULTR2;2::GCaMP3$ and $pATML1::G-CaMP3$ responded slowly to $Z$-3-HAL (Fig. 6f, g).

To better understand GLV propagation pathways, we observed $[Ca^{2+}]_{cyt}$ changes at the cellular level using an upright confocal laser

**Fig. 4 | Z-3-HAL and E-2-HAL trigger the leaf surface potential changes and defense-related gene expression in Arabidopsis.** **a, b** Simultaneous recording of changes in the leaf surface potential and $[Ca^{2+}]_{cyt}$ (yellow arrowheads) in the tip region of L1 after the application of Z-3-HAL (**a**) and E-2-HAL (**b**) solutions 5 mm from the tip of L1. The chemical solution was applied in a plastic tube indicated by a white dashed line (0 s). The white arrow indicates the recording electrode. Scale bar, 2.5 mm. **c, d** Quantification of the leaf surface potential and $[Ca^{2+}]_{cyt}$ signatures induced by Z-3-HAL (**c**) and E-2-HAL (**d**). An enlarged graph is presented on the right. Error bars, mean ± SE. N = 10 biologically independent samples.
**e** Comparison of the time points at which the change in the leaf surface potential

and $Ca^{2+}$ signal was detected. The change in signal used to measure velocity was defined as an increase or a decrease to above 2 standard deviation (SD) of the pre-stimulation levels. An asterisk denotes a significant difference based on two-tailed Student's t-test (*, $P < 0.05$). Error bars, mean ± SE. N = 10 replicates per line.
**f** Transcript levels of HSP90.1, ZAT12, OPR3, and JAZ7 in L1 of Arabidopsis at 30 and 60 min after treatment with Z-3-HAL, E-2-HAL, or DMSO (Mock). ACT8 was used as an internal reference for standardization. Error bars, mean ± SE. N = 5. Different letters denote significant differences based on one-way ANOVA followed by Bonferroni's post hoc test ($P < 0.05$).

scanning microscope (Fig. 7a–c). A rapid $[Ca^{2+}]_{cyt}$ increase was first observed in guard cells (pGC1::GCaMP3) within 1 min of Z-3-HAL exposure (Fig. 7d, g, h, and Supplementary Movie 12). An increase in $[Ca^{2+}]_{cyt}$ was detected earlier in mesophyll cells (pRBCS1A::GCaMP3) than in epidermal cells (pATML1::GCaMP3) (Fig. 7e–h and Supplementary Movies 13–14). These observations were consistent with the results obtained using Arabidopsis expressing GCaMP3 driven by the 35S promoter (Supplementary Fig. 6 and Movie 15). Based on these findings, we hypothesized that plant GLV sensory transduction could be initiated by GLV flux into inner tissues via stomata, resulting in subsequent defense signaling activation.

### Stomata play a critical role in rapid Z-3-HAL perception

To examine the function of stomata in rapid GLV sensory transduction, we used the phytohormone abscisic acid (ABA) to induce stomatal closure[40], as well as stomatal mutants exhibiting abnormal stomatal movement phenotypes[41–43]. Pretreatment of Arabidopsis WT leaves with ABA resulted in stomatal closure (Supplementary Fig. 7), and the Z-3-HAL-induced increase in $[Ca^{2+}]_{cyt}$ was delayed compared with that in leaves pretreated with a stomatal opening buffer solution (Mock, Fig. 8). Loss-of-function mutants of SLOW ANION CHANNEL-ASSOCIATED 1 (slac1) and OPEN STOMATA 1 (ost1) exhibit impaired stomatal closure in the presence of ABA[41,42] (Supplementary Fig. 7). Z-3-HAL-induced $[Ca^{2+}]_{cyt}$ signatures in ABA-treated slac1-2 and ost1-3 mutants expressing GCaMP3 driven by the 35S promoter were similar to those in Mock-treated leaves (Fig. 8). Moreover, a critical role of stomata in rapid Z-3-HAL-induced electrical signaling was confirmed via the time-course analysis of leaf surface potential changes using ABA-pretreated leaves (Supplementary Fig. 8). These results support our hypothesis that GLV uptake from the atmosphere into tissues via stomata is the main pathway of rapid GLV sensory transduction in plants, which in turn activates plant defense responses.

## Discussion

We visualized $Ca^{2+}$ signal transduction in intact Arabidopsis plants exposed to VOCs emitted by mechanically- and herbivore-damaged plants (Figs. 1 and 2) and found that Z-3-HAL and E-2-HAL function as airborne signaling molecules triggering $[Ca^{2+}]_{cyt}$ increases, electrical signals, and transcriptional changes in Arabidopsis leaves (Figs. 3, 4). Since VOC components other than these two C6 aldehydes were unable to elicit $[Ca^{2+}]_{cyt}$ changes in Arabidopsis (Fig. 3f), the aldehyde moiety in their structures appeared necessary for activating $[Ca^{2+}]_{cyt}$-based defense signaling. E-2-HAL containing an α,β-unsaturated carbonyl moiety is a reactive electrophile species that causes cell damage because of its ability to form adducts with nucleophiles[44]. Z-3-HAL, which has a β,γ-unsaturated carbonyl group, exerts physiological effects on various organisms[14,20]. Surprisingly, n-HAL and Z-3-HAC failed to trigger $[Ca^{2+}]_{cyt}$ increases (Fig. 3f), whereas $Ca^{2+}$ response to Z-3-HAC exposure was previously reported in tomatoes[29]. These findings suggest that plants possess species-specific sophisticated VOC recognition systems in addition to a mechanism capable of recognizing structural differences among GLV components (such as specialized receptor proteins)[12,45]. Indeed, some volatile-specific receptors in plants have been isolated, including those for ethylene[46] and β-caryophyllene[25].

In addition to the volatile-specific perception system, VOC uptake pathways involving metabolic processes have been proposed to act as other primary pathways for downstream VOC signaling activation[22]. Z-3-HOL accumulated in tomato cells is glycosylated, leading to enhanced resistance against S. litura via the insecticidal effects[47]. However, considering that neither Z-3-HOL nor β-caryophyllene triggered $[Ca^{2+}]_{cyt}$ changes (Fig. 3f), plants might be equipped with specialized VOC perception systems that activate downstream signaling independently at different levels (for example, $Ca^{2+}$ signals, transcriptional changes, and metabolism).

The C6 aldehyde group can prime/induce plant defense responses to abiotic stresses[20]. For example, heat and photooxidative stresses elicit enhanced E-2-HAL production in Arabidopsis, tomatoes[21,48], and tobacco[49]. GLV-exposed Arabidopsis seedlings exhibit enhanced heat tolerance[21]. In this study, oxidative stress- and heat-responsive genes were upregulated upon GLV exposure in a $Ca^{2+}$-dependent manner (Fig. 4f and Supplementary Fig. 3b). Although the detailed molecular mechanism underlying stress-related signaling activation upon GLV exposure remains unclear, $[Ca^{2+}]_{cyt}$ increases could mediate both stress-responsive gene expression[31,50] and immediate GLV formation via $Ca^{2+}$ binding to the PLAT domain in LOX, a key enzyme for GLV biosynthesis[51]. Different experimental approaches, such as transcriptome analysis, could provide new insights into the mechanism underlying enhanced stress tolerance through GLV-induced $Ca^{2+}$ signals.

The local concentration of Z-3-HAL at a distance of 5 mm from the 3.0 M solution, which consistently induced stable $[Ca^{2+}]_{cyt}$ increases, was estimated at 6.7 mM (Supplementary Fig. 5). Similarly, 381 nmol Z-3-HAL was detected within 10 min of exposure to 0.5 g homogenized Arabidopsis leaf tissues (Supplementary Fig. 2), indicating that receiver Arabidopsis is exposed to approximately 6.6 mM Z-3-HAL in 30 s. Although the homogenization of Arabidopsis leaf tissues might be unrealistic in nature, these estimations indicate that the concentration of Z-3-HAL solution corresponds to that emitted by plants and that the experimental conditions used in this study are relevant to the potential exposure or emission of Z-3-HAL in a natural context.

Plants possess the ability to efficiently absorb a wide range of surrounding atmospheric VOCs and accumulate them into their tissues. For example, tomato can absorb a significant amount of atmospheric methacrolein, with estimates ranging from 33% to 41% of the total methacrolein content present in the air[52]. The minimum amount of Z-3-HAL required to induce detectable $Ca^{2+}$ signals was 0.1 M (Fig. 5a). Considering that approximately 12.8 nmol Z-3-HAL was adsorbed by the adsorbents within 30 s of exposure to 0.1 M Z-3-HAL, Arabidopsis accumulates approximately 4.2 nmol Z-3-HAL at 33% of the VOC-adsorbing capacity of the adsorbent. Based on the previous calculation[34], a single injury (7.5 µg, 0.05 mm²) produces 12.9 and 2.9 pmol Z-3-HAL in tomatoes and Arabidopsis, respectively (for detail, see Supplementary Fig. 2 and Methods). If these estimations are accurate, the Z-3-HAL emitted following a single injury is unlikely to induce detectable $Ca^{2+}$ signals. To induce the release of 12.8 nmol Z-3-HAL (the accumulation of 4.2 nmol Z-3-HAL in Arabidopsis leaves) and $Ca^{2+}$ signals under our experimental conditions, approximately 148.8 (992.3 mm²) and 662.1 (4409.0 mm²) mg of leaves need to be injured in

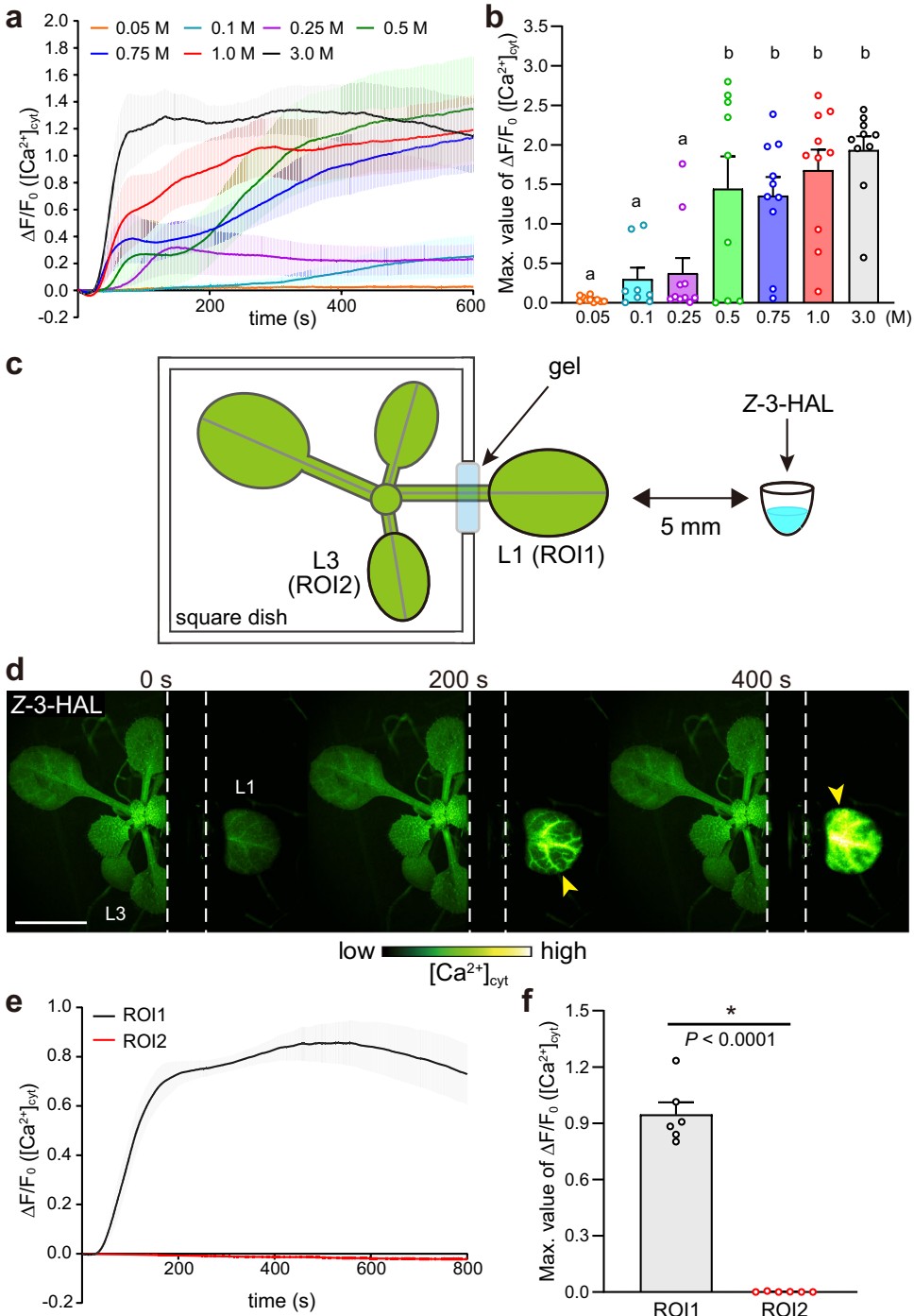

**Fig. 5 | *Z*-3-HAL-induced Ca²⁺ signature occurs locally and in a concentration-dependent manner. a** Concentration dependency of the effects of *Z*-3-HAL activity on the $[Ca^{2+}]_{cyt}$ signature after exposure. Error bars, mean ± SE. $N = 10$ biologically independent samples, except for 0.1 M ($N = 8$) and 0.5 M ($N = 9$). **b** Comparison of the maximal $[Ca^{2+}]_{cyt}$ changes detected in receiver *Arabidopsis* upon exposure to each concentration of *Z*-3-HAL. Error bars, mean ± SE. $N = 10$ biologically independent samples, except for 0.1 M ($N = 8$) and 0.5 M ($N = 9$). Different letters denote significant differences based on one-way ANOVA followed by Tukey's honestly significant difference post hoc test ($P < 0.05$). **c** Diagram presenting *Arabidopsis* and the ROIs used to analyze $[Ca^{2+}]_{cyt}$ changes.

L1 was spatially segregated using a square dish. ROI1 and ROI2 were set at local (L1) and systemic (L3) regions, respectively. **d** The time-course changes in $[Ca^{2+}]_{cyt}$ (yellow arrowheads) in L1 and L3 leaves after exposing L1 to *Z*-3-HAL. The dashed outline indicates the position of the square dish. Scale bar, 5 mm. **e** Quantification of $[Ca^{2+}]_{cyt}$ signatures in L1 (ROI1) and L3 (ROI2). Error bars, mean ± SE. $N = 6$ biologically independent samples. **f** Comparison of the maximal $[Ca^{2+}]_{cyt}$ changes detected in L1 (ROI1) and L3 (ROI2) of receiver *Arabidopsis* upon exposure to 3.0 M *Z*-3-HAL. Error bars, mean ± SE. $N = 6$ biologically independent samples. An asterisk denotes statistically significant differences based on two-tailed Student's *t* test (*, $P < 0.05$).

tomatoes and *Arabidopsis*, respectively. This size of agricultural damage might be realistic in nature[53].

Although it is unlikely that plants are continuously exposed to high GLV concentrations (e.g., 3.0 M *Z*-3-HAL) under natural conditions, it should be noted that GLVs do not easily diffuse because of their high molecular weight, which may lead to high local GLV concentrations around damaged plants[54,55]. Considering these findings, the possibility that plant cells and tissues are temporarily exposed

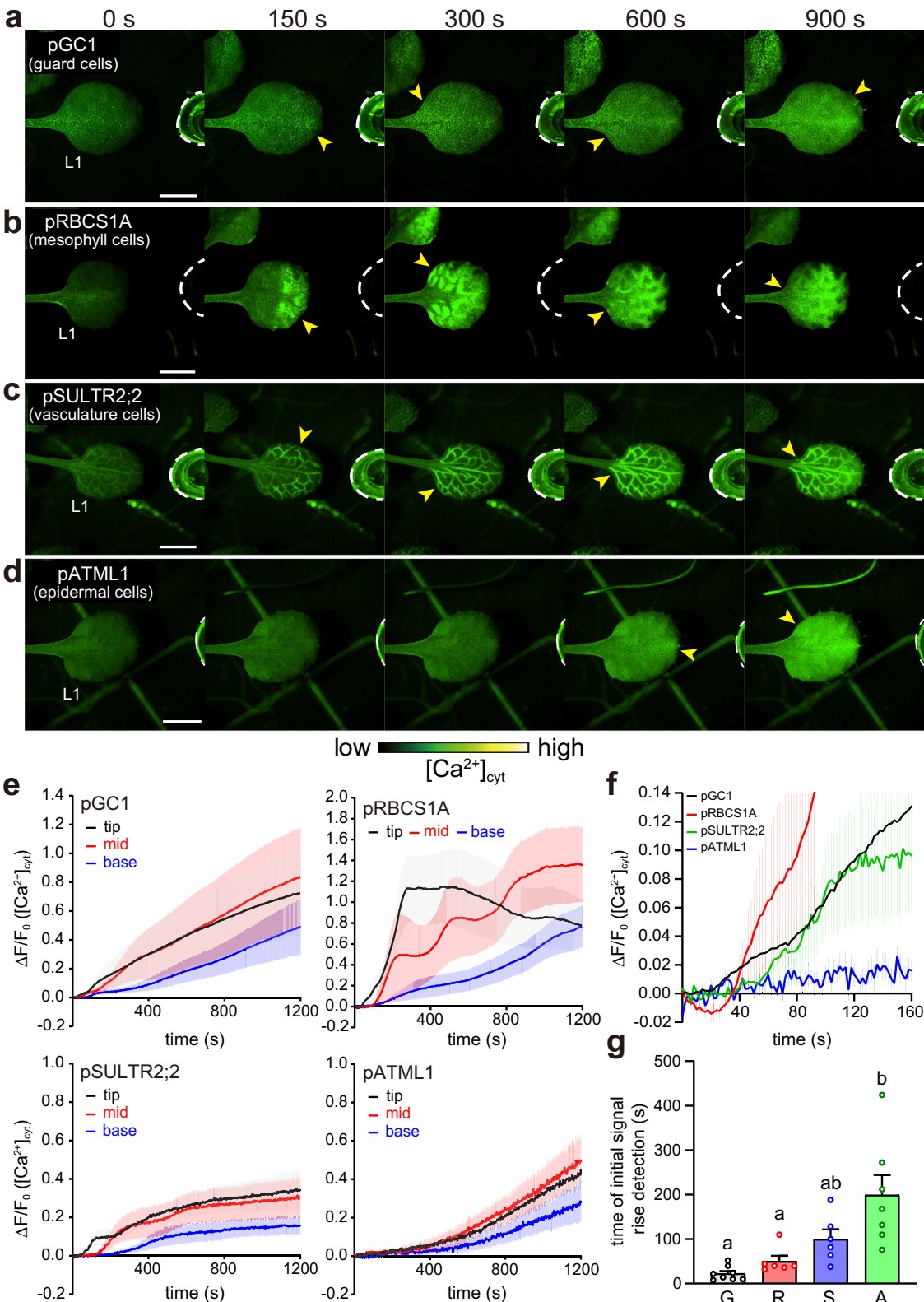

**Fig. 6 | [Ca²⁺]cyt increases in guard, mesophyll, and vasculature cells, and subsequently in epidermal cells upon Z-3-HAL exposure. a–d** Snapshots of Ca²⁺ changes (yellow arrowheads) in L1 of *Arabidopsis* expressing GCaMP3 driven by the tissue-specific promoters GC1 (**a**), RBCS1A (**b**), SULTR2;2 (**c**), and ATML1 (**d**) after *Z*-3-HAL exposure. Scale bar, 2.5 mm. **e** Quantification of [Ca²⁺]cyt signatures in the tip, mid, and base regions. Error bars, mean ± SE. *N* = 6 biologically independent samples for pRBCS1A and pSULTR2;2, *N* = 8 for pGC1 and *N* = 7 for pATML1. **f** Enlarged graph displaying the onset of [Ca²⁺]cyt increases within 160 s in the tip region of each

transgenic line. **g** Comparison of the time points at which the increase in the initial Ca²⁺ signal was detected in the tip region of L1 of *pGC1::GCaMP3* (G), *pRBCS1A::G-CaMP3* (R), *pSULTR2;2::GCaMP3* (S), and *pATML1::GCaMP3* (A), following *Z*-3-HAL exposure. The increase in signal used to calculate velocity was defined as an increase to above 2 SD of the pre-stimulation levels. Error bars, mean ± SE. *N* = 6 biologically independent samples for pRBCS1A and pSULTR2;2, *N* = 8 for pGC1 and *N* = 7 for pATML1. Different letters denote statistically significant differences based on one-way ANOVA followed by Tukey's honestly significant difference post hoc test (*P* < 0.05).

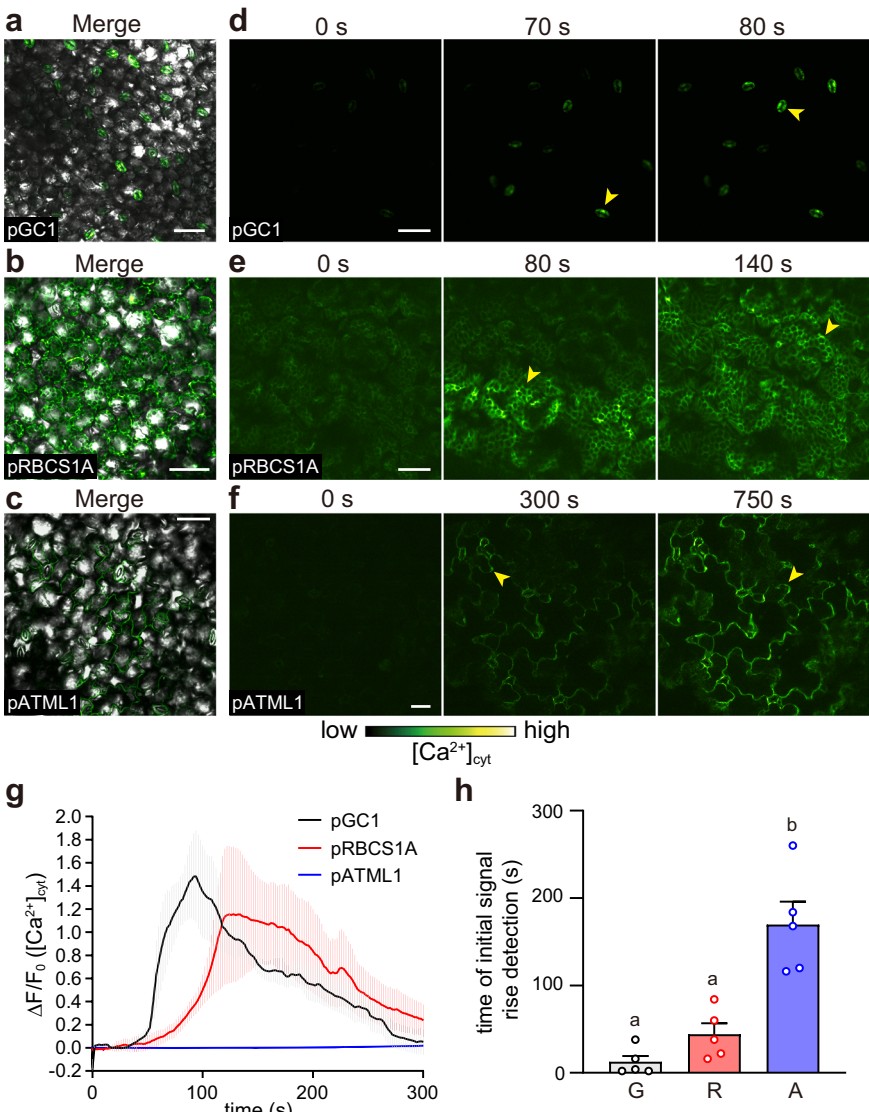

**Fig. 7 | Upright confocal laser scanning microscopy reveals instant *Z*-3-HAL-induced [Ca²⁺]$_{cyt}$ increases in guard cells, followed by mesophyll cells.**
**a**–**c** Localization of the GCaMP3 signal for *pGC1::GCaMP3* in guard cells (**a**), for *pRBCS1A::GCaMP3* in mesophyll cells (**b**), and for *pATML1::GCaMP3* in epidermal cells (**c**). The merged GFP signal and bright field images were presented as an overlay. **d**–**f** Snapshots of Ca²⁺ changes (yellow arrowheads) of [Ca²⁺]$_{cyt}$ levels in L1 of *pGC1::GCaMP3* (**d**), *pRBCS1A::GCaMP3* (**e**), and *pATML1::GCaMP3* (**f**) after exposure to *Z*-3-HAL. Scale bar, 50 µm. **g** Quantification of the [Ca²⁺]$_{cyt}$ levels in the L1 of each transgenic line. Error bars, mean ± SE. $N = 5$ biologically independent samples. **h** Comparison of the time points at which the increase in the Ca²⁺ signal was detected in *pGC1::GCaMP3* (G), *pRBCS1A::GCaMP3* (R), and *pATML1::GCaMP3* (A) following *Z*-3-HAL exposure. The increase in signal used to calculate velocity was defined as an increase to above 2 SD of the pre-stimulation levels. Error bars, mean ± SE. $N = 5$ biologically independent samples. Different letters denote statistically significant differences based on one-way ANOVA followed by Tukey's honestly significant difference post hoc test ($P < 0.05$).

to high GLV concentrations under specific circumstances (e.g., when receiver plants are in close proximity to disrupted plants capable of emitting numerous GLVs, including *Vigna radiata* and *Momordica charantia*[56], in response to herbivory in nature) cannot be dismissed. In fact, we detected Ca²⁺ signals in response to VOCs released from plants consumed by herbivores (Fig. 1). To further clarify this phenomenon, it would be beneficial to employ advanced technologies, such as the real-time detection of atmospheric VOC concentrations, and precisely determine the VOC adsorption capacity and efficiency of plants[57].

Using real-time Ca²⁺ imaging combined with pharmacological and genetic approaches, we proposed a model for the spatiotemporal propagation pathways of VOCs in plants (Fig. 9) where stomata play a critical role in perceiving VOC cues. Some studies support our idea of the importance of stomata for VOC uptake, especially for the absorption of atmospheric gases such as $CO_2$, and air pollutants[23,58].

For example, VOC uptake by plant tissues is efficiently facilitated when stomata are opened[23]. Aldehyde compounds can also be absorbed into the leaf interior via stomata[58]. Conversely, the delayed Ca²⁺ signals in epidermal cells could be explained by the presence of the cuticle, which functions as a permeability barrier. This idea is supported by a previous finding that O₃ deposition in cuticles was negligibly small[59]. Taken together, it is possible that stomata serve as a plant gateway mediating rapid VOC entry into interspaces in tissues.

Two glutamate receptor-like genes (GLRs) that are localized in phloem and xylem contact cells in the vasculature are activated in response to wounds, resulting in the propagation of Ca²⁺ and electrical signals from the wound site to distant organs[31,60]. Although *Z*-3-HAL-induced [Ca²⁺]$_{cyt}$ increases were detected in the vasculature tissue (Fig. 6c, e), no Ca²⁺ signal propagation toward systemic leaves was observed (Fig. 5d). Given that the amplitude and propagation rate of

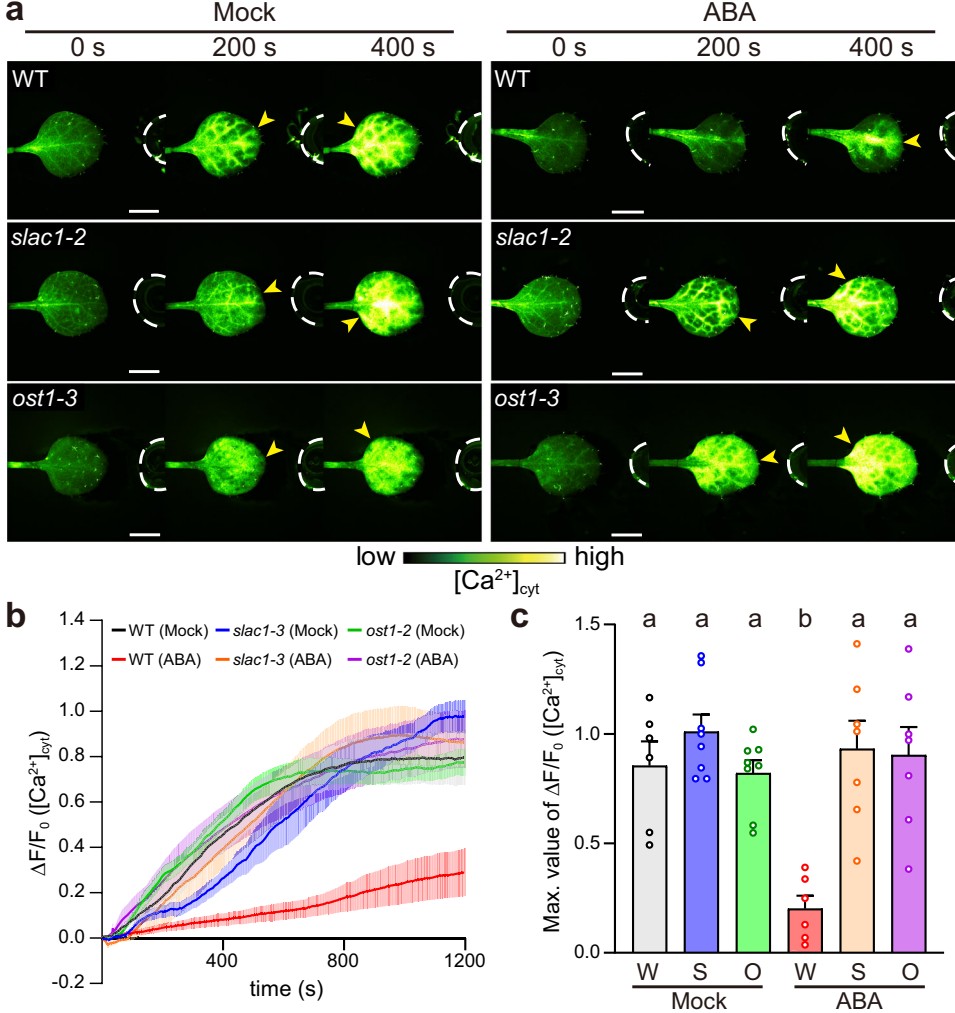

**Fig. 8 | Stomatal opening plays a crucial role in $[Ca^{2+}]_{cyt}$ increases upon *Z*-3-HAL exposure. a** Pretreatment with ABA (20 μM) delayed $[Ca^{2+}]_{cyt}$ increases (yellow arrowheads) induced by exposure to *Z*-3-HAL in WT leaves but not the leaves of the *slac1-2* and *ost1-3* mutants expressing GCaMP3 driven by the 35S promoter. Scale bar, 2.5 mm. **b** Quantification of *Z*-3-HAL-induced $[Ca^{2+}]_{cyt}$ signatures in the whole region of the detached leaf. Error bars, mean ± SE. *N* = 6 biologically independent samples for WT (Mock) and WT (ABA), *N* = 7 for *slac1-2* (Mock), *slac1-2* (ABA) and *ost1-3* (ABA), and *N* = 8 for *ost1-3* (Mock). **c** Comparison of the maximal $[Ca^{2+}]_{cyt}$ changes detected in the detached leaves of WT (W), *slac1-2* (S), and *ost1-3* (O). Error bars, mean ± SE. *N* = 6 biologically independent samples for WT (Mock) and WT (ABA), *N* = 7 for *slac1-2* (Mock), *slac1-2* (ABA) and *ost1-3* (ABA), and *N* = 8 for *ost1-3* (Mock). Different letters denote statistically significant differences based on one-way ANOVA followed by Tukey's honestly significant difference post hoc test ($P < 0.05$).

the wound-triggered systemic signals depend on the type and intensity of the stimuli[61], and the necessity of damaging the main vein of the leaf[62], GLVs might not activate the key elements required for long-distance signal transmission, such as GLRs, or the local response might not reach the threshold required to allow the $Ca^{2+}$/electrical signals to move out of the local leaf. Interestingly, changes in the leaf surface potential preceded changes in $[Ca^{2+}]_{cyt}$ upon GLV exposure (Fig. 4e), as systemic electrical signals preceded $Ca^{2+}$ signals upon wounding[63,64]. Membrane depolarization could be induced by the activation of ion channels, such as $Cl^-$-permeable, ROS-sensitive, or ligand-gated channels, followed by $[Ca^{2+}]_{cyt}$ increases via $Ca^{2+}$ influx and efflux through plasma- and endo-membranes, respectively[63,64].

The wide-field real-time imaging approach used in this study provided physiological insights with significantly higher resolution using intact *Arabidopsis*, revealing the details of $Ca^{2+}$ signals in response to GLVs in a more comprehensive manner. This methodology allowed the assessment of spatial and temporal aspects of GLV perception pathways at the cellular level. Additionally, by integrating real-time imaging with other techniques, we achieved a deeper understanding of the comprehensive orchestration of GLV responses,

including $Ca^{2+}$ signals and other signaling mechanisms such as electrical signals. This approach can be further extended to investigate VOC signaling networks across plant taxa using mutants that are defective in the putative elements of VOC responses. Furthermore, this $Ca^{2+}$ imaging method can serve as a robust tool for investigating the molecular basis of airborne plant signaling, both within (e.g., *Arabidopsis* to *Arabidopsis*) and between species (e.g., tomato to *Arabidopsis*) (Figs. 1 and 2).

## Methods

### Plant material and growth condition

The seeds of *Arabidopsis thaliana* (accessions Col-0, No-0 and *hpl1* mutant in the Ler-0 background[34]) were surface-sterilized and sown on sterile Murashige and Skoog (MS) agar medium [1× MS salts, 1% (w/v) sucrose, 0.01% (w/v) myoinositol, 0.05% (w/v) MES, and 0.5% (w/v) gellan gum; pH 5.7 adjusted with 1 N KOH]. After incubation in the dark at 4 °C for 2 days, the plates were placed horizontally at 22 °C in a growth chamber under continuous light (90–100 μmol/m²/s) for approximately 2 weeks before use. After 2 weeks, *Arabidopsis* leaves were numbered from oldest to youngest[62]. Two-week-old *Arabidopsis*

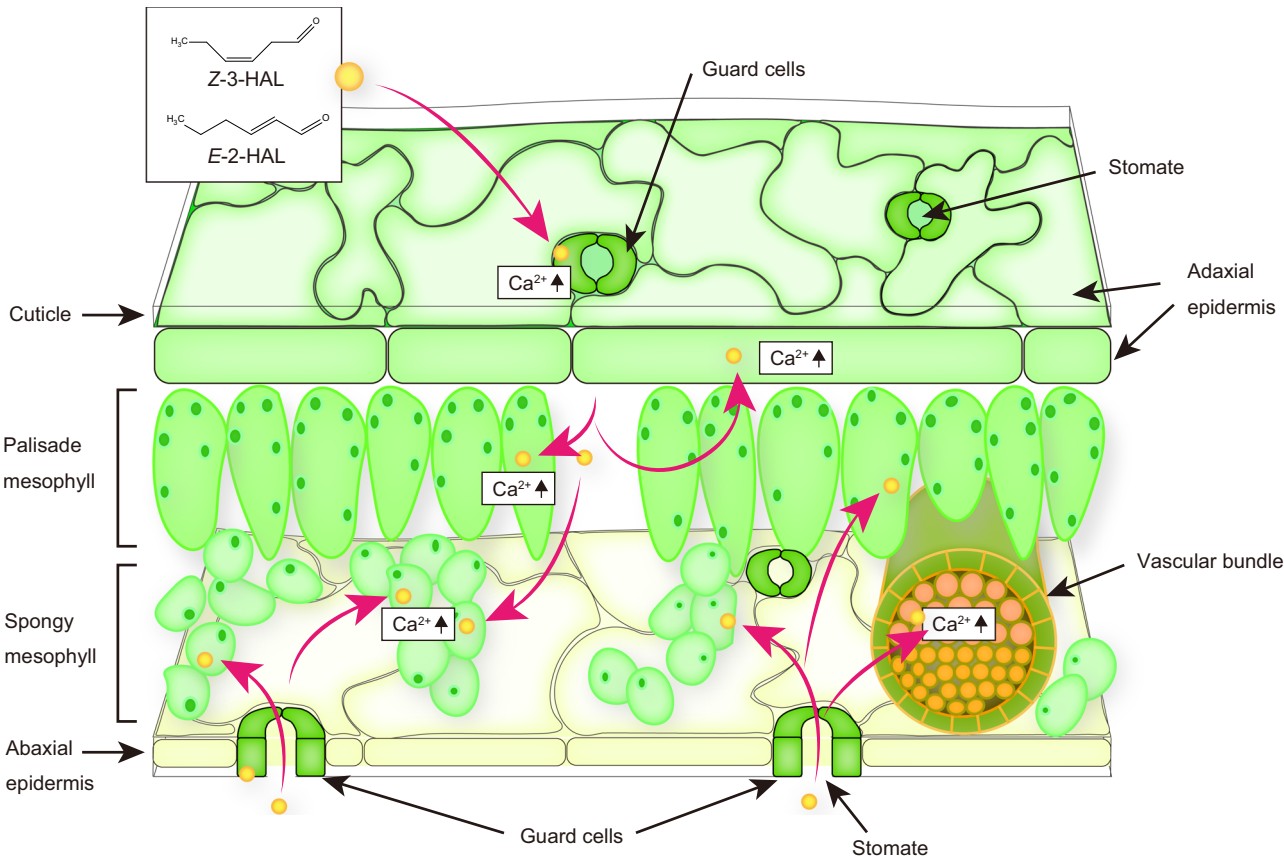

**Fig. 9 | Volatile C6 aldehyde sensory transduction system in *Arabidopsis*.**
Volatile C6 aldehydes, such as *Z*-3-HAL and *E*-2-HAL, in the atmosphere are initially perceived by guard cells, which leads to $[Ca^{2+}]_{cyt}$ increases. C6 aldehydes can enter the interspace of leaf tissues via stomata, leading to the subsequent activation of defense responses in mesophyll and vasculature cells. $[Ca^{2+}]_{cyt}$ increases are delayed in epidermal cells because of the presence of physical barriers, such as cuticles.

(accession No-0 and Ler-0) plants were transferred to soil and grown at 22 °C in a growth chamber under short-day conditions (8-h light/16-h dark photoperiod, 90–100 µmol/m²/s) for approximately 2 months before use. Two-week-old (accession Col-0) plants were used for $Ca^{2+}$ imaging. Two-month-old plants (accession No-0 and Ler-0) were used for preparation of homogenized leaf tissues and leaves that were subjected to feeding by *S. litura*. Surface-sterilized tomato (*Solanum lycopersicum* cv. Micro-Tom) seeds were incubated overnight at 4 °C and sown on MS agar medium. Two-week-old plants were transferred to soil and grown at 25 °C in a growth chamber under long-day conditions (14-h light/10-h dark photoperiod, 90–100 µmol/m²/s) for approximately 2 months before use. Tomato (*Solanum lycopersicum* cv. Minicarol) seedlings potted in soil were purchased from a home improvement store located in Saitama, Japan. These seedlings were cultivated in a temperature-controlled room under long-day conditions (14-h light/10-h dark photoperiod, 90–180 µmol/m²/s) at 25–29 °C for more than 12 days and then used to prepare leaves that were consumed by *S. litura*.

**Insects**
Eggs of *S. litura* (Fabricius) (Lepidoptera: Noctuidae) were purchased from Sumika Technoservice Co. Ltd. (https://www.chemtex.co.jp). The insects were reared on a homemade artificial diet in the laboratory at 25 °C. Fifth-instar *S. litura* larvae were used for $Ca^{2+}$ imaging.

**DNA cloning and transformation**
For the GCaMP3 constructs used for tissue-specific expression, genomic sequences of the 5′ end of the open-reading frames of *GC1* (1716 bp, At1g22690), *RBCS1A* (1976 bp, At1g67090), *SULTR2;2* (2001 bp, At1g77990), and *ATML1* (3378 bp, At4g21750) were amplified via polymerase chain reaction (PCR) targeting restriction enzyme sites from *Arabidopsis* genomic DNA and used as the promoter sequences. These fragments were digested and inserted into the corresponding sites of the pAN19 vector containing the GCaMP3 and nopaline synthase terminator (NOSt) sequence, resulting in construction of the vectors *pGC1::GCaMP3 NOSt*, *pRBCS1A::GCaMP3 NOSt*, *pSULTR2;2::G-CaMP3 NOSt*, and *pATML1::GCaMP3 NOSt*. The entire cassettes of GCaMP3 sequences driven by tissue-specific promoters were isolated via *Not*I digestion and cloned into the *Not*I site of the plant binary vector pBIN42. All binary vectors were transformed into the *Agrobacterium tumefaciens* strain GV3101 via electroporation. *Arabidopsis* plants were transformed using the floral dip method[65]. To establish *Arabidopsis* mutants expressing GCaMP3, the entire cassette of *p35ss::GCaMP3 NOSt*[31] was transformed into the *slac1-2* and *ost1-3* mutants. The primers used for cloning are detailed in Supplementary Table 1.

**Real-time $[Ca^{2+}]_{cyt}$ imaging**
Real-time $[Ca^{2+}]_{cyt}$ imaging of the entire plants was performed as described previously[31]. For $Ca^{2+}$ imaging at the cellular level, GCaMP3 signals were acquired using an upright confocal laser scanning microscope (A1R, Nikon). GCaMP3 was excited using a 488-nm laser/488-nm dichroic mirror, and fluorescent signals were detected at 510–560 nm using the GaAsP detector of the microscope. NIS-Elements imaging software was used to analyze GCaMP3 signals over time at several regions of interest (ROI). To calculate fractional fluorescence changes (ΔF/F), the following equation was used: $\Delta F/F = (F-F_0)/F_0$, where $F_0$ denotes the average baseline fluorescence determined by

the average of $F$ over the first 10 frames of the recording before treatment.

### Volatile treatment

β-Pinene and *n*-HAL were purchased from Wako Pure Chemical Industries, Ltd. α-pinene and β-caryophyllene were obtained from Tokyo Chemical Industry Co., Ltd. MeJA, *E*-2-HAL, *Z*-3-HOL, and *Z*-3-HAC were purchased from Sigma-Aldrich Co. LLC. *Z*-3-HAL was obtained from Nihon Zeon Co., Ltd. Each chemical was dissolved in dimethyl sulfoxide (DMSO) to make a stock solution of 3.0 M. To examine the concentration dependency of the responses, *Z*-3-HAL was dissolved in DMSO at concentrations of 0.05, 0.1, 0.25, 0.5, 0.75, and 1.0 M. A 0.2-mL plastic tube cut with a scissor at a height of 5 mm was placed 5 mm from the tip region of L1, and then 10 µl of each volatile solution was applied.

To selectively treat L1 with *Z*-3-HAL, a hole was created on the side of a square plastic dish with a cutter, and L1 was spatially segregated from other parts of the plant by inserting it into the hole, as described in Fig. 5c. Then, the hole was filled with 0.2% agarose gel, and Ca²⁺ imaging was performed.

For treatment with VOCs emitted by homogenized plants, approximately 10 and 5 g of the aboveground parts of 2-month-old *Arabidopsis* and tomato, respectively, were excised with a scissor and homogenized with a pestle in a mortar. Disrupted leaf tissues were immediately transferred to 1.5-mL plastic tubes and placed in close proximity to receiver *Arabidopsis* expressing GCaMP3. Subsequently, the MS agar plate was closed with a clear plastic cover to increase the volatile concentration, and then Ca²⁺ imaging was performed.

To facilitate exposure to VOCs emitted from leaves consumed by *S. litura*, we established an experimental setup consisting of two plastic bottles, flow meters (NFM-V-P-A-1, TEKHNE Corp.), and air pump (HD-603, FEDOUR), as shown in Fig. 1a. Briefly, approximately 7 g of the aboveground parts of 2-month-old *Arabidopsis* and tomato plants was excised with scissors. The severed leaves and 50–75 fifth instar *S. litura* larvae were placed inside a plastic bottle (approximately 215 cm³). The bottle was sealed with parafilm to prevent VOC leakage and incubated for 30 min. Prior to the experiment, air was pumped into an empty plastic bottle at a rate of approximately 450 mL/min and was directed toward receiver *Arabidopsis* for 10 min through a cotton-filled tip. This step was crucial for facilitating the adaptation of plants to the experimental conditions. Following adaptation, exposure to VOCs emitted by leaves consumed by *S. litura* as well as Ca²⁺ imaging was initiated by connecting the bottle and switching the valve. The air was pumped into the bottle at a rate of approximately 200 mL/min.

### Recording of surface potential

Surface potential changes were measured as described previously with minor modifications[66]. Ag/AgCl recording electrodes with a diameter of 0.2 mm were prepared via chloridation with hypochlorous acid and fixed to electrode holders. The recording electrodes were fixed to the tip region of L1 of *Arabidopsis* plants with the application of 5-µL droplets of 10 mM KCl. The handmade Ag/AgCl wire with a diameter of 0.5 mm was used as a reference electrode and inserted into the agar growth medium. For measuring surface potential changes, an operational amplifier (Axopatch 200A, Axon Instruments), headstage amplifier (CV-201A headstage, Axon Instruments), digitizer (Digidata 1322A, Axon Instruments), and electrophysiology data acquisition software (Clampex 9.2, Axon Instruments) were used. Simultaneous measurements were performed using the SMZ25 microscope in a Faraday cage. Surface potential changes were sampled at 5 kHz. To compare the timing of the initial signal change with Ca²⁺ increases, each data point was subsequently extracted at a reduced frequency of 0.5 Hz. For calculating the surface potential changes (ΔV), the following equation was used: $\Delta V = V - V_0$, where $V$ denotes the potential difference (PD) between the recording and the reference electrodes at a

certain time and $V_0$ denotes the averaged baseline PD determined by the mean of $V$ over the first 10 frames of the recording before treatment.

### ABA treatment

L1 of *Arabidopsis* plants was harvested from 2-week-old plants, floated in stomatal opening buffer solution (5 mM MES and 50 mM KCl; pH 6.1 adjusted with 1 N KOH) abaxial side down, and incubated under light (90–100 µmol/m² /s) for 2 h to open the stomata. Subsequently, ABA (20 µM) (Sigma-Aldrich) was added to the solution to induce stomatal closure. After 2 h of treatment, leaves were used for the subsequent experiments. For Ca²⁺ imaging and surface potential recording, detached leaves were transferred to MS agar medium and treated with 3.0 M *Z*-3-HAL as previously described. For stomatal aperture measurement, epidermal strips were prepared by peeling away the epidermal cell layer using a clear Scotch tape as described previously[67]. The adaxial leaf surface was fixed with cover glass, and stomata were observed under an upright confocal laser scanning microscope (A1R, Nikon). The stomatal aperture was measured using NIS-Elements imaging software (Nikon).

### Pharmacological treatment

LaCl₃·7H₂O (Wako) and EGTA (Dojindo Laboratories) were dissolved in liquid MS medium at a final concentration of 50 mM. Eight-day-old *Arabidopsis* seedlings were transferred into a Petri dish filled with liquid MS medium containing either 50 mM LaCl₃ or EGTA and incubated for 16 h prior to experiments. For the washout assay, *Arabidopsis* seedlings that had been treated with 50 mM LaCl₃ or EGTA were uprooted from MS medium and washed 5 times with liquid MS medium devoid of these inhibitors to remove any residual inhibitors. Subsequently, the seedlings were transferred to an inhibitor-free MS agar plate and incubated for 24 h before conducting Ca²⁺ imaging.

### Total RNA isolation, cDNA synthesis, and quantitative PCR

To extract total RNA, L1 and aboveground parts were harvested from 2-week-old *Arabidopsis* plants and 8-day-old *Arabidopsis* seedlings, respectively (Fig. 4f and Supplementary Fig. 3b). The harvested tissues were immediately frozen using liquid nitrogen. Total RNA was extracted from flash-frozen leaf tissue using the Plant Total RNA Mini Kit (FAVORGEN) following the manufacturer's instructions. The samples were further treated with RNase-free DNase I to remove any residual genomic DNA using the RNase-Free DNase Set (QIAGEN) according to the manufacturer's instructions. First-strand cDNA was then synthesized from the total RNA (500 ng) in a 10-µL reaction (50 ng of total RNA/µL) with PrimeScript™ RT Master Mix (Perfect Real-Time) for RT-PCR (Takara). In a 96-well optical PCR plate (ABgene), cDNA proportional to 10 ng of starting total RNA was combined with 100 nM of each primer (Supplementary Table 1) and 7.5 µL of 2× Brilliant III Ultra-Fast SYBR Green QPCR Master Mix (Agilent Technologies) to a final volume of 15 µL. Using *Arabidopsis ACT8* as an internal reference for standardization, qPCR was performed using the CFX96 Touch Deep Well Real-Time PCR System as well as CFX Maestro Software (Bio-Rad) with the following cycling parameters: 95 °C for 3 min; 40 cycles of 95 °C for 5 s and 60 °C for 10 s; and 1 cycle of dissociation from 65 °C to 95 °C with 0.5 °C increments. The expression of the marker genes was quantified using the quantification cycle [Cq].

### Volatile analysis

VOCs emitted from homogenized leaves were identified and quantified as described previously with some modifications[68]. Briefly, the aboveground parts of *Arabidopsis* (No-0 and Ler-0 accession) and tomato (*Solanum lycopersicum* cv. Micro-Tom) plants were excised with a razor blade and weighed. Five hundred milligrams of tissues were immediately placed into a mortar and homogenized using a pestle and liquid nitrogen. Then, these tissues were transferred to a

grass vial (22 ml, Perkin Elmer). A MonoTrap cartridge (silica monolith matrix coated with octadecyl silyl group and activated carbon, RGPS TD, GL Sciences) was suspended in the headspace of the glass vial, allowing it to adsorb VOCs for 10 min. To quantify the volatilization of Z-3-HAL from DMSO solution, the MonoTrap cartridge was placed 5 mm from a 0.2-mL plastic tube containing 10 μL of 0.03 M Z-3-HAL solution in DMSO, and VOC adsorption was performed for 0.5, 1, 2 and 5 min (Supplementary Fig. 5). Volatiles collected by the cartridge were analyzed by a GC–MS system (GCMS QP2030, Shimadzu) equipped with a thermal desorption system (TQ8040-NX, Shimadzu). Volatiles were desorbed at 250 °C for 10 min with He gas flow (70 mL/min) and concentrated onto a trap set at −25 °C. Volatiles were desorbed again at 250 °C for 2 min and fractionated with a DB-WAX capillary column (30 m × 0.25 mm, 0.25 μm film thickness, Agilent). The GC oven program was maintained at an initial temperature of 40 °C (held for 5 min), followed by a ramp of 5.0 °C/min to a final temperature of 200 °C (held for 2 min). The electron ionization mode with an ionization voltage of 70 eV was used, and the $m/z$ was recorded from 40 to 400. To identify each compound, we used the retention indices and MS profiles of the corresponding authentic specimens. To construct calibration curves for Z-3-HAL and E-2-HAL, a given amount of authentic compounds (generous gift from Zeon Co., and purchased from Fujifilm Wako Pure Chemicals, respectively) was directly injected onto the MonoTrap cartridge and analyzed as previously described.

To estimate Z-3-HAL production from a single injury, we followed a previously described method[34]. This estimation was based on the detection of Z-3-HAL that elicits $[Ca^{2+}]_{cyt}$ increases in *Arabidopsis* following leaf homogenization as described in Supplementary Fig. 2. The total amounts of Z-3-HAL produced by homogenized *Arabidopsis* and tomato leaves were quantified as 380.9 and 1718.7 nmol/gFW, respectively. These values represent the maximum capacity of Z-3-HAL synthesis by the leaves. Based on previous research[34], the average weight of a leaf was defined as 150 μg/mm². Considering the estimated area of a single injury to be 0.05 mm² as previously described[34], we calculated that a single wound would result in the production of approximately 2.9 and 12.9 pmol of Z-3-HAL for *Arabidopsis* and tomatoes, respectively.

### Statistical analysis

We performed Student's *t* test for pairwise analysis and one-way analysis of variance followed by Bonferroni's or Tukey's post hoc tests using GraphPad Prism (GraphPad Software, Inc.) to compare multiple samples. Statistical significance was indicated by $P < 0.05$. All data are presented as the mean ± SE.

### Reporting summary

Further information on research design is available in the Nature Portfolio Reporting Summary linked to this article.

## Data availability

The data that support the findings of this study are available from the corresponding author upon reasonable request. Source data are provided with this paper.

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

## Acknowledgements
The authors thank Juntaro Negi (Kyushu University) for providing *slac1-2* and *ost1-3* mutant seeds. This research was supported by grants from KAKENHI (18H05491) to M.T., (22KJ0451) to T.U., and by grants from A-STEP and Shiraishi Foundation of Science Development to M.T.

## Author contributions
Y.A., T.U., T.H., and M.T. designed the study. Y.A., T.U., and T.H. performed Ca$^{2+}$ imaging, electrophysiological measurements, qPCR analysis. K.M. performed volatile analysis. Y.A., T.U., T.H., and K.M. performed data analysis. Y.A., T.U., and M.T. wrote the manuscript. All authors discussed the results and contributed to the manuscript.

## Competing interests
The authors declare no competing interests.
