## [Peer Review File · Nature Communications]

REVIEWER COMMENTS

Reviewer #1 (Remarks to the Author):

The work entitled "Green leaf volatile sensory calcium transduction in Arabidopsis" by Aratani et al. reports that "volatile organic compounds (VOCs)" released from damaged leaves of Arabidopsis (ecotype No-0) and *S. lycopersicum* induce calcium transients in the rosette leaves of Col-0 Arabidopsis plants expressing the genetically encoded calcium sensor GCaMP3.

The first interesting observation is that the VOCs released by homogenized tomato leaves can induce calcium transients of greater intensity than those of Arabidopsis. The authors then suggest that among the various possible VOCs released by homogenized leaves the most active ones are (Z)-3-Hexenal and (E)-2-Hexenal which alone induce calcium transients in leaf cells.

It must be said that this result is not new as it was reported previously by Maffei's team working with tomato plants using calcium-sensitive dyes instead of genetically encoded sensors (Zebelo, et al., 2012 Plant Science 196, 93-100). In addition to the demonstration that these two VOCs induce calcium transients both Aratani et al. that in Zebelo et al. provide evidence that they induce depolarization of the leaf surface.

The most interesting novelty presented in this work is the tissue-specific analysis of the response to VOCs by employing Arabidopsis plants in which the GCaMP3 sensor is under the control of different promoters. From the analysis done, the authors suggest that stomata guard cells are the first cell type that responds to VOCs, followed by mesophyll cells. Furthermore, the treatment of plants with the hormone ABA or the analysis of mutants impaired in the stomatal closure suggests that the open state of the stomata influences the cytosolic Ca²⁺ transient triggered in response to VOCs.

The work presents interesting ideas and certainly from a technical point of view it is innovative. In particular, the possibility of analyzing the responses of different tissues in adult and unmanipulated plants (thanks to the use of a stereomicroscope) is a powerful approach but not new.

Having said that, I believe that the presentation of the data is somehow poor and a little bit superficial. In many cases, there is a lack of in-depth analysis of the data which then somehow limits the authors to discuss them in an appropriate manner. In my specific comments, I provide some suggestions on how to improve the presentation of the data, their analysis, etc. Importantly, a better statistical analysis is required.

In conclusion, although I believe that the work presents some interesting ideas and is technically sound, it is also true that some of the data had already been reported about ten years ago, which somewhat limits the novelty of the work. In addition, as the same authors honestly report in the discussion, it is not certain that their experimental design represents a true physiological condition. The concentrations of the VOCs used are in fact very high, and even the homogenization of the leaves itself is a condition that is difficult to achieve in nature. An experiment that the authors could carry out would be to test whether the sole wounding of a leaf is able to induce in neighbouring plants calcium transients in plants that are not subjected to any stress.

Specific comments

-Figure 1a is somehow misleading. The *A. thaliana* (No-0) and *S. lycopersicum* names on the left side seem to indicate that the plants shown in the images belong to these two species. I would use Col-0 on the left side of the images and indicate the homogenized leaves, and sources of VOCs, under the first and second rows of images.

Moreover, a more detailed analysis of the traces should be presented. I would suggest reporting at least the maximum DF/F0 and also performing a statistical analysis.

Maybe, it would be also useful to cite in the legend that to perform the experiment same amount of homogenized material for both species was used.

-L91. It would be important to report here that both No-0 and *S. lycopersicum* homogenized leaves are indeed able to produce the two different tested VOC compounds.

-L98-99. I would suggest the authors to better present these data. If they want to compare the effects of the two compounds, aside from presenting the maximum calcium peaks, I suggest comparing the propagation speed as well as the response appearance to the different VOCs which seem to be delayed in response to (E)-2-Hexenal compared to (Z)-3-Hexenal.

I would suggest showing arrowheads with different colours for the three different leaf regions analyzed (tip, mid, and base). The colours should be chosen accordingly to the traces shown in panels b and d.

-L111. Instead of writing "Vm depolarization" I would suggest specifying that they are measuring the leaf surface potential.

-L113-114. I would suggest the authors show a close-up of panels 3b and 3d corresponding to the time at which the cytosolic Ca²⁺ increase and the surface potential depolarization occur. Particularly, in Figure 3d it seems that in response to (E)-2-Hexenal the cytosolic Ca²⁺ increase anticipates the depolarization, whereas the opposite occurs in response to (Z)-3-Hexenal. Providing a statistical analysis by comparing the time at which the two events occur will clarify this point.

-L128. I think, aside from Figure 4a, it is important to show box plots where the comparison of the Ca²⁺ peaks observed in response to the different concentrations is reported. Moreover, I would also like to see a comparison of the times at which the Ca²⁺ peaks are observed. Surely, a proper statistical analysis must be presented.

-L128. What concentration of (Z)-3-Hexenal was used? This information is reported only in the M&M and it seems they used 0.25 M. This is strange because in Figure 4d treatment with 0.25 M induces a transient where the DF/F0 reaches 1.0, where the same concentration in Figure 4a triggers only a DF/F0 of 0.3. Moreover, can the authors explain which was the rationale for using 0.25 M among the other concentrations tested? Theoretically, also for the investigation of systemic responses, a dose-dependent analysis would be useful.

-L144. Also, in this case, I would suggest the authors to better analyze and discuss the data. I would not only show Ca²⁺ traces, but I would compare the times at which the Ca²⁺ increase starts in the different sensor lines. Also comparing the maximum Ca²⁺ peaks for the different regions of the different lines would be useful. Showing the data in this way would help to better discuss the results. Also, the choice of the selected images at different time points is weird. I would select the same time points (e.g. 0, 150, 300, 600, and 900 s)

for all lines.

-L147. I have the same comments made for Figure 5. A better analysis of the data is needed. As a minor comment, there is a need to indicate that the Overlay is between the GFP and bright field channels.

-L157. I would introduce at the beginning of the paragraph the rationale behind the use of ABA and mutants impaired in ABA-induced stomatal closure. I would also highlight in the text that for this experiment, the GCaMP3 was under the control of the CaMV35S promoter. As for the other figures, a box plot for the comparison of the Ca²⁺ peaks should be presented.

-L326. 16 hrs treatment with 50 mM of La³⁺ or EGTA is a very strong condition, potentially harmful for plants. Maybe, lower concentrations should be tested.

-L382. It is important to report in the Figure legend that Arabidopsis ACT8 was used as an internal reference for qRT-PCR standardization.

Supplementary Figure 3. In this case, the response to (E)-2-Hexenal is stronger than the (Z)-3-Hexenal a result that is opposite to what is shown in Figure 3. The authors need to comment on this.

Supplementary Figure 4. I would suggest moving this panel to the main Figure 3 and performing a statistical analysis by comparing the times at which they observe the cytosolic Ca²⁺ increase in the different lines.

Reviewer #2 (Remarks to the Author):

In this manuscript Aratani et al. report their investigation of an interesting aspect of plant-to-plant communication, which is the triggering of defense reactions by volatile organic compounds (VOCs) in plants. Specifically, the authors used wide-field real-time imaging of GCaMP3-expressing plants to clarify the role of Ca²⁺ signaling and its spatio-temporal dynamics in plant-to-plant communication. They report that VOCs emitted from homogenized leaves induce local Ca²⁺ signals in exposed plants. Moreover, they confirm that (Z)-3-Hexenal and (E)-2-Hexenal induce Ca²⁺ signals and membrane depolarization. Finally, by using plant lines that express Ca²⁺ reporter proteins under control of cell-type specific promoters, the authors provide evidence that stomata indeed provide entrance points for VOCs that elicit these Ca²⁺ signals. Overall, the manuscript is well written, and the experiments are principally well executed and documented. I personally like the movies very much. I have the following major comments:

- The authors use crushed leaf homogenates to induce Ca²⁺ signals in the exposed plants. This technical setup falls short to really establish that such a plant-to-plant communication (that evokes Ca²⁺ signals in the receiving plant) indeed occurs in nature. To uphold this claim, it would require to use an experimental setup as it has been reported for example in ref. 26 (Zebelo et al., 2012; Figure 1). Here, an airflow was directed over a plant, on which caterpillars were feeding and then subsequently directed to the plant that was analyzed.
- The result that VOC exposure triggers Ca²⁺ signals in exposed plants is not really novel

and has principally been reported by Zebelo et al., 2012 and (with aequorin plants) by Asai et al., 2009. Admittedly, the data that are provided here are of much higher quality and resolution.

- Also, (Z)-3-Hexenal and (E)-2-Hexenal have been intensively studied in the past and have already previously been reported to be the compounds that trigger Ca²⁺ signals and membrane depolarization (Zebelo et al., 2012).

- The authors detail here that guard cells represent the entry points of VOCs and that consequently, VOC-triggered Ca²⁺ signals start around these leaf valves. These are now very convincing data. Nevertheless, with more indirect evidence a role of guard cell conductance for VOC uptake has been indicated before.

Reviewer #3 (Remarks to the Author):

The response of plants to VOC is known since long time and the effects of green leaf volatiles on calcium variations have already been demonstrated in other plant species. Therefore, the results here presented do not represent a novelty and confirm previous studies. The only further information given here is the sequential activity of calcium release and the possible determination of guard cells as the first VOC sensing cells.

The work is of significance in the general response of plants to airborne VOCs because it provides further evidence of the involvement of calcium in plant-insect and plant-environment interactions. However, with regards to the specific literature it lacks one important information, the quantitative and qualitative determination of VOC emitted by the homogenized material here used. This is also affecting all results obtained with the presumed molecules and the unrealistic concentrations used. Without the chemical characterization of the emitted VOCs the data here presented lack of realism, because hundred of VOCs are emitted by plants. In fact, by comparing the presented results with those of the literature (see for instance <https://doi.org/10.1074/jbc.ra118.005843>) the concentrations used are about 1000 times as expected. Nevertheless, for many other technical aspects the work is original.

Due to the above criticism (above all the lack of any chemical characterization and quantification) many of the conclusions are unsustainable and before publication additional experiments are need in order to assess: 1) the exact chemical composition and the quantification of VOCs released from homogenized tissues (both Arabidopsis and Tomato). 2) The quantification of how much of these VOCs are volatilized in the experimental area (very often the authors write "in close proximity", this description means nothing in terms of repeatability of the experiment).

The methodology related to the detection of Calcium variations is sound and, specifically to the calcium signaling area, the work meets the expected standards in the plant science field. The methods are appropriate.

Other issues:

- the authors report experiments with beta-pinene (Line 92) but no results are reported for this monoterpene

- The paragraph of lines 111-115 is the repetition of previous results and should be deleted since it does not represent a novelty.

- lines 122-123: this is already well known form the literature

Line 133 Supplementary video 5 shows the opposite of what reported in this sentence

Line 173. The authors have measured differences in V_m I'm not sure they can claim they showed electrical signals.

Lines 174-175. The authors did not provide a full genome response to GLVs, therefore

based only on 4 genes analyzed they cannot claim “diverse Ca²⁺-dependent defense responses”

The sentence of line 177-178 contradicts what is reported in lines 182-187. The authors did not know the true composition of the emitted VOCs therefore they do not know whether other molecules are present. Moreover, the generalization to aldehyde moiety is only supported by unrealistic concentrations, while it is known that plants respond differently to low and very low concentrations of VOCs. Indeed there are several mono and sesquiterpene aldehydes and these are known not to induce defense in plants.

Lines 206-207. The authors did not provide any evidence that stomata “activate plant defense responses”, they are probably involved but surely not the only one involved. With the presented data this sentence is pure speculation.

Lines 212-215. The authors did not provide any evidence of an oxidative stress or heat stress related to VOC perception. They simply tested a couple of genes and speculated over the full process of responses to ROS and heat.

Line 216. The authors evidently ignore that Calcium signals are known to be key components in VOC signaling since tens of years.

Lines 234-237. At last the authors realized that without the careful quantitative and qualitative characterization of the VOC emitted any consideration (especially when brought by unrealistic concentrations) remains purely speculative.

Based on the above consideration, the manuscript cannot be accepted for publication.

Reply to Reviewers.

We greatly appreciate the Editor and Reviewers for their encouraging comments and thoughtful reviews, which helped in significantly improving our manuscript. We have addressed all the raised points, and the specific responses are provided in a point-by-point manner.

Reviewer #1:

The work entitled "Green leaf volatile sensory calcium transduction in Arabidopsis" by Aratani et al. reports that "volatile organic compounds (VOCs)" released from damaged leaves of Arabidopsis (ecotype No-0) and S. lycopersicum induce calcium transients in the rosette leaves of Col-0 Arabidopsis plants expressing the genetically encoded calcium sensor GCaMP3.

The first interesting observation is that the VOCs released by homogenized tomato leaves can induce calcium transients of greater intensity than those of Arabidopsis. The authors then suggest that among the various possible VOCs released by homogenized leaves the most active ones are (Z)-3-Hexenal and (E)-2-Hexenal which alone induce calcium transients in leaf cells.

It must be said that this result is not new as it was reported previously by Maffei's team working with tomato plants using calcium-sensitive dyes instead of genetically encoded sensors (Zebelo, et al., 2012 Plant Science 196, 93-100). In addition to the demonstration that these two VOCs induce calcium transients both Aratani et al. that in Zebelo et al. provide evidence that they induce depolarization of the leaf surface.

The most interesting novelty presented in this work is the tissue-specific analysis of the response to VOCs by employing Arabidopsis plants in which the GCaMP3 sensor is under the control of different promoters. From the analysis done, the authors suggest that stomata guard cells are the first cell type that responds to VOCs, followed by mesophyll cells. Furthermore, the treatment of plants with the hormone ABA or the analysis of mutants impaired in the stomatal closure suggests that the open state of the stomata influences the cytosolic Ca²⁺ transient triggered in response to VOCs.

The work presents interesting ideas and certainly from a technical point of view it is innovative. In particular, the possibility of analyzing the responses of different tissues in adult and unmanipulated plants (thanks to the use of a stereo microscope) is a powerful approach but not new.

Having said that, I believe that the presentation of the data is somehow poor and a little bit superficial. In many cases, there is a lack of in-depth analysis of the data which then somehow limits the authors to discuss them in an appropriate manner. In my specific comments, I provide some suggestions on how to improve the presentation of the data, their analysis, etc. Importantly, a better statistical analysis is required.

In conclusion, although I believe that the work presents some interesting ideas and is technically sound, it is also true that some of the data had already been reported about ten years ago, which somewhat limits the novelty of the work. In addition, as the same authors honestly report in the discussion, it is not certain that their experimental design represents a true physiological condition. The concentrations of the VOCs used are in fact very high, and

even the homogenization of the leaves itself is a condition that is difficult to achieve in nature. An experiment that the authors could carry out would be to test whether the sole wounding of a leaf is able to induce in neighbouring plants calcium transients in plants that are not subjected to any stress.

We sincerely appreciate your valuable comments and suggestions, which have helped strengthen our manuscript. We acknowledge that the previous pioneering study (Zebelo et al., Plant Science 2012) had revealed the involvement of Ca^{2+} signaling in plant VOC perception systems. However, only single snapshots of Ca^{2+} imaging in tomatoes were provided in the previous article. As the fluorescence dye (calcium orange) was extracellularly applied to tomato leaves, some technical limitations must be addressed. (1) As most fluorescent dyes are charged, they cannot permeate the plasma membrane (Gilroy et al. Plant Cell 1991). (2) Although the previous study appeared to use a neutral (acetoxymethyl ester) form of the dye (calcium orange-AM), there are abundant enzymes esterases in the extracellular (apoplastic) region, potentially rendering the dye impermeable to the plasma membrane (Zhang et al. Plant Journal 1998). (3) Intracellular fluorescence dyes lacking dextrans can be transported from the cytosol to vacuoles. (4) Finally, the cell type that absorbs the dyes and emits fluorescent signals remains unclear.

To overcome these limitations, we established highly sensitive wide-field stereomicroscope and confocal microscope systems in combination with molecular genetic techniques, enabling real-time, high-resolution, tissue-specific Ca^{2+} imaging in intact *Arabidopsis* plants upon GLV exposure. We also combined the electrophysiological and pharmacological techniques with Ca^{2+} imaging to simultaneously analyze Ca^{2+} and electrical signals. These advanced technologies provide spatial and temporal information regarding plant VOC perception at the organ and cellular levels and add new insights into plant-to-plant communication. We have highlighted the novelty of our study in this field in the Discussion section and emphasized the advancements in this study.

As you suggested, we statistically analyzed our data to determine the differences between samples. Based on these analyses, we obtained robust evidence for significant differences in our data. We have added detailed explanations of the statistical analyses in the responses to specific comments.

We also greatly appreciate your valuable suggestion regarding the experimental designs. To assess the potential occurrence of this phenomenon under realistic natural conditions, we conducted additional experiments to (1) observe Ca^{2+} signatures in receiver *Arabidopsis* exposed to volatiles released from plants fed by the generalist herbivore *Spodoptera litura* and (2) quantify the VOCs emitted from tomato and *Arabidopsis* leaves as well as GLV solutions in collaboration with Dr. Kenji Matsui (Yamaguchi University, Japan), a plant biologist working on GLVs. We found that exposure to VOCs released by tomato and *Arabidopsis* leaves fed on by *S. litura* induced the obvious $[\text{Ca}^{2+}]_{\text{cyt}}$ increases in receiver *Arabidopsis*. We believe that the Ca^{2+} responses fit in the range of natural conditions, although practically high GLV concentrations are required to consistently observe stable $[\text{Ca}^{2+}]_{\text{cyt}}$ increases in *Arabidopsis*. We have added movies and descriptions of the results in the revised manuscript.

Revised main text: Page 3: First, we monitored $[Ca^{2+}]_{\text{cyt}}$ increases in *Arabidopsis* (receiver) following exposure to VOCs emitted from *Arabidopsis* plants (source of VOCs) fed on by the common cutworm (*Spodoptera litura*; Fig. 1a, b and Supplementary Video 1). This Ca^{2+} signal was rapidly transmitted to all parts of the plant within 20 min (Fig. 1c). Similar results were obtained when receiver *Arabidopsis* was exposed to VOCs released by tomato leaves (source of VOCs) consumed by *S. litura* (Fig. 1b, c and Supplementary Video 2).

Revised main text: Page 8: Approximately 3.9 nmol Z-3-HAL was adsorbed by adsorbents from a 0.03 M Z-3-HAL solution within 30 s (Supplementary Fig. 5). Considering the volume of the adsorbents (58 μL), it can be estimated that the local concentration of Z-3-HAL at a distance of 5 mm from the 3 M solution, which consistently induced stable $[Ca^{2+}]_{\text{cyt}}$ increases, was 6.7 mM. Similarly, 381 nmol Z-3-HAL was detected within 10 min of exposure to 1 g homogenized *Arabidopsis* leaf tissues (Supplementary Fig. 2), indicating that receiver *Arabidopsis* is exposed to approximately 6.6 mM Z-3-HAL in 30 s, which fits in the range of the concentration emitted from the 3 M Z-3-HAL solution. Although the homogenization of *Arabidopsis* leaf tissues might be unrealistic in nature, these estimations indicate that the concentration of Z-3-HAL solution corresponds to that emitted by plants and that the experimental conditions used in this study are relevant to the potential exposure or emission of Z-3-HAL in a natural context.

Revised main text: Page 8: Based on the calculation methodology described in a previous study³¹, a single injury (7.5 μg , 0.05 mm^2) produces 12.9 and 2.9 pmol Z-3-HAL in tomatoes and *Arabidopsis*, respectively (for detail, see Supplementary Fig. 2 and Methods). If these estimations are accurate, then the Z-3-HAL emitted following a single injury are unlikely to induce detectable Ca^{2+} signals. To induce the release of 12.8 nmol Z-3-HAL (the accumulation of 4.2 nmol Z-3-HAL in *Arabidopsis* leaves) and Ca^{2+} signals under our experimental conditions, approximately 148.8 (31.5 mm \times 31.5 mm = 992.3 mm^2) and 662.1 (66.4 mm \times 66.4 mm = 4409.0 mm^2) mg of leaves need to be injured in tomatoes and *Arabidopsis*, respectively. This size of agricultural damage might be realistic in nature⁵⁷.

Revised main text: Page 10: To facilitate exposure to VOCs emitted from leaves consumed by *S. litura*, we established an experimental setup consisting of two plastic bottles, flow meters (NFM-V-P-A-1, TEKHNE Corp.), and air pump (HD-603, FEDOUR), as shown in Fig. 1a. Briefly, approximately 7 g of the aboveground parts of 2-month-old *Arabidopsis* and tomato plants was excised with scissors. The severed leaves and 50–75 fifth instar *S. litura* larvae were placed inside a plastic bottle (approximately 215 cm^3). The bottle was sealed with parafilm to prevent VOC leakage and incubated for 30 min. Prior to the experiment, air was pumped into an empty plastic bottle at a rate of approximately 450 mL/min and was directed toward receiver

Arabidopsis for 10 min through a cotton-filled tip. This step was crucial for facilitating the adaptation of plants to the experimental conditions. Following adaptation, exposure to VOCs emitted by leaves consumed by *S. litura* as well as Ca^{2+} imaging was initiated by connecting the bottle and switching the valve. The air was pumped into the bottle at a rate of approximately 200 mL/min.

Revised main text: Page 13: To estimate Z-3-HAL production from a single injury, we followed a previously described method³¹. This estimation was based on the detection of Z-3-HAL that elicits $[\text{Ca}^{2+}]_{\text{cyt}}$ increases in *Arabidopsis* following leaf homogenization as described in Supplementary Fig. 2. The total amounts of Z-3-HAL produced by homogenized *Arabidopsis* and tomato leaves were quantified as 380.9 and 1718.7 nmol/gFW, respectively. These values represent the maximum capacity of Z-3-HAL synthesis by the leaves. Based on previous research³¹, the average weight of a leaf was defined as 150 $\mu\text{g}/\text{mm}^2$. Considering the estimated area of a single injury to be 0.05 mm^2 as previously described³¹, we calculated that a single wound would result in the production of approximately 2.9 and 12.9 pmol of Z-3-HAL for *Arabidopsis* and tomatoes, respectively.

Revised main text: Page 15: Fig. 1. Exposure to VOCs emitted by homogenized leaves and leaves consumed by the herbivore *S. litura* induces $[\text{Ca}^{2+}]_{\text{cyt}}$ increases in receiver *Arabidopsis* leaves.

(a) The experimental setup for Ca^{2+} imaging in *Arabidopsis* (receiver) upon exposure to VOCs emitted by leaves consumed by *S. litura* larvae is schematically illustrated. Prior to the experiment, the receiver *Arabidopsis* in a plastic dish was acclimated by directing airflow from an empty plastic bottle for 10 min, allowing its adaptation to the experimental conditions. Subsequently, receiver *Arabidopsis* was exposed to VOCs emitted from a plastic bottle containing *S. litura* larvae and either *Arabidopsis* or tomato leaves (source of VOCs) by connecting the bottle and manipulating the valve. The black arrow indicates the direction of airflow. (b) Changes in $[\text{Ca}^{2+}]_{\text{cyt}}$ (yellow arrowheads) in *Arabidopsis* expressing GCaMP3 in response to VOCs released from *Arabidopsis* (upper) and tomato (*Solanum lycopersicum*) leaves (below) consumed by *S. litura* larvae. White dashed lines indicate the position of the tip of the tube from where the airflow emerges. Scale bar, 5 mm. (c) Quantification of $[\text{Ca}^{2+}]_{\text{cyt}}$ signatures in leaf 1 (L1). Error bars, mean \pm SE. $N > 3$. (d) The experimental setup for Ca^{2+} imaging in *Arabidopsis* (receiver) upon exposure to VOCs from homogenized *Arabidopsis* or tomato leaves (source of VOCs) is schematically illustrated. In total, 10 g of *Arabidopsis* leaves or 5 g of tomato leaves was homogenized with liquid nitrogen using a mortar and pestle. The resulting homogenized tissues were immediately transferred to 1.5-mL plastic tubes. Subsequently, Ca^{2+} imaging was initiated by placing the tubes in close proximity to receiver *Arabidopsis*.

Specific comments

-Figure 1a is somehow misleading. The *A. thaliana* (No-0) and *S. lycopersicum* names on the left side seem to indicate that the plants shown in the images belong to these two species. I would use *Col-0* on the left side of the images and indicate the homogenized leaves, and sources of VOCs, under the first and second rows of images. Moreover, a more detailed analysis of the traces should be presented. I would suggest reporting at least the maximum $\Delta F/F_0$ and also performing a statistical analysis. Maybe, it would be also useful to cite in the legend that to perform the experiment same amount of homogenized material for both species was used.

We apologize for the confusion. As requested, we have redesigned Fig. 1 to clarify our experimental design. First, we have added a schematic diagram of the experiment and defined the sample types (“source of VOCs” and “receiver”) in Fig. 1d. We also provided explanations of the sample types on the left and upper sides of the images in Fig. 1e. Second, we determined the maximal $\Delta F/F_0$ for tomato and *Arabidopsis* data, and the increase in $[Ca^{2+}]_{\text{cyt}}$ induced by tomato VOCs was significantly higher than that induced by VOCs emitted by *Arabidopsis* (Fig. 1g). Furthermore, we concluded a similar analysis comparing the maximum $\Delta F/F_0$ for WT *Arabidopsis* and the *hpl1* mutant. We found that the Ca^{2+} signature induced by WT VOCs was significantly stronger than that induced by *hpl1* VOCs (Supplementary Fig. 1c). We have added the data and descriptions in the revised text.

Revised main text: Page 15: (d) The experimental setup for Ca^{2+} imaging in *Arabidopsis* (receiver) upon exposure to VOCs from homogenized *Arabidopsis* or tomato leaves (source of VOCs) is schematically illustrated. In total, 10 g of *Arabidopsis* leaves or 5 g of tomato leaves was homogenized with liquid nitrogen using a mortar and pestle. The resulting homogenized tissues were immediately transferred to 1.5-mL plastic tubes. Subsequently, Ca^{2+} imaging was initiated by placing the tubes in close proximity to receiver *Arabidopsis*.

(g) Comparison of the maximal $[Ca^{2+}]_{\text{cyt}}$ changes detected in receiver *Arabidopsis* upon exposure to VOCs emitted by homogenized *Arabidopsis* (*A.t*) or tomato (*S.l*) leaves. An asterisk denotes statistically significant differences based on Student’s *t*-test (*, $P < 0.05$).

Revised main text: Page 3: In a time-course analysis of $[Ca^{2+}]_{\text{cyt}}$ changes in leaf 1 (L1), we found that VOCs emitted by homogenized tomato leaves caused larger signal changes than those emitted by *Arabidopsis* (Fig. 1f, g). These results suggest that VOCs emitted by damaged *Arabidopsis* and tomato plants caused $[Ca^{2+}]_{\text{cyt}}$ changes in neighboring intact *Arabidopsis* plants.

We apologize for the insufficient explanation regarding the experimental conditions. For the homogenization experiment, we used 5 g of tomato leaves and 10 g of *Arabidopsis* leaves. We have added an additional description regarding this specific condition in the legend of Fig. 1d.

Revised main text: Page 15: In total, 10 g of *Arabidopsis* leaves or 5 g of tomato leaves was homogenized with liquid nitrogen using a mortar and pestle. The resulting homogenized tissues were immediately transferred to 1.5-mL plastic tubes.

-L91. It would be important to report here that both No-0 and S. lycopersicum homogenized leaves are indeed able to produce the two different tested VOC compounds.

This is an excellent point, and we have conducted GC–MS analysis to measure the amount of VOCs emitted by the homogenized leaves of WT *Arabidopsis* and tomato. Our results confirmed the rapid and substantial emission of both Z-3-HAL and E-2-HAL after homogenization. Moreover, the *hpl1* mutant exhibited lower ability to emit GLVs upon homogenization than the WT. We have added this information in the revised Results section.

Revised main text: Page 4: Using gas chromatography mass spectrometry (GC–MS) and MonoTrap RGPS TD, which is a high-quality adsorbent, we further analyzed the VOC components rapidly produced by homogenized tomato and *Arabidopsis* leaves. Tomato leaves emitted significantly higher amounts of Z-3-HAL and E-2-HAL than wild-type (WT) *Arabidopsis* leaves upon homogenization (Supplementary Fig. 2). Moreover, the levels of these compounds emitted by homogenized *hpl1* leaves were remarkably low (<25% of that emitted by WT leaves; Supplementary Fig. 2), consistent with the results of Ca²⁺ signature levels.

-L98-99. I would suggest the authors to better present these data. If they want to compare the effects of the two compounds, aside from presenting the maximum calcium peaks, I suggest comparing the propagation speed as well as the response appearance to the different VOCs which seem to be delayed in response to (E)-2-Hexenal compared to (Z)-3-Hexenal. I would suggest showing arrowheads with different colours for the three different leaf regions analyzed (tip, mid, and base). The colours should be chosen accordingly to the traces shown in panels b and d.

We thank you for the valuable suggestions, which have helped us improve the quality of our manuscript. We have added results regarding the comparison of the maximum peaks and velocities passing through each ROI. This analysis indicated that Ca²⁺ signals induced by Z-3-HAL were faster than those induced by E-2-HAL, as indicated by the reviewer. Furthermore, we have added dashed circles in different colors in the images to highlight the specific points corresponding to each ROI (tip, mid, and base) and help reveal the experimental data visually. The text has been revised to reflect these points.

Revised main text: Page 3: By measuring the velocities of Z-3-HAL- and E-2-HAL-induced Ca²⁺ transmission, we revealed that the signal transmission induced by Z-3-HAL (0.24–0.30 mm/s; N = 10) was faster than that induced by E-2-HAL (0.01–0.02 mm/s; N = 10; Fig. 2e).

Revised main text: Page 17: (b) Time-course changes in $[Ca^{2+}]_{cyt}$ (yellow arrowheads) in the L1 of *Arabidopsis* after applying Z-3-HAL (upper) and E-2-HAL (lower) at a distance of 5 mm from the tip of L1. The chemical solution was applied into a plastic tube, indicated by an orange dashed line (0 s). Dashed white, red, and blue circles indicate the position of the tip, mid, and base regions, respectively. Scale bar, 2.5 mm.

(d) Comparison of the maximal $[Ca^{2+}]_{cyt}$ changes detected in each ROI upon exposure to Z-3-HAL and E-2-HAL. An asterisk denotes statistically significant differences based on Student's *t*-test (*, $P < 0.05$). **(e)** Velocities (mm/s) of the Ca^{2+} signals transmitted between the tip and mid regions (V_1) and between the mid and base regions (V_2) induced by Z-3-HAL and E-2-HAL were analyzed. An asterisk denotes statistically significant differences based on Student's *t*-test (*, $P < 0.05$).

-L111. Instead of writing "Vm depolarization" I would suggest specifying that they are measuring the leaf surface potential.

As indicated, we have revised and replaced the term "Vm depolarization" with "leaf surface potential" throughout the manuscript.

-L113-114. I would suggest the authors show a close-up of panels 3b and 3d corresponding to the time at which the cytosolic Ca^{2+} increase and the surface potential depolarization occur. Particularly, in Figure 3d it seems that in response to (E)-2-Hexenal the cytosolic Ca^{2+} increase anticipates the depolarization, whereas the opposite occurs in response to (Z)-3-Hexenal. Providing a statistical analysis by comparing the time at which the two events occur will clarify this point.

This is another excellent point. As requested, we have included the additional enlarged data specifically representing the onset of both electrical and Ca^{2+} signals induced by GLVs on the right side of Fig. 3c–d. Interestingly, by analyzing the timing of the initial detectable signal changes, we revealed that the surface potential change significantly preceded the onset of the Ca^{2+} signal in the plant GLV transduction system. Accordingly, we have revised the manuscript to reflect these points.

Revised main text: Page 4: Z-3-HAL and E-2-HAL exposure in *Arabidopsis* leaves resulted in rapid changes in the leaf surface potential, which is spatiotemporally coupled with changes in $[Ca^{2+}]_{cyt}$ (Fig. 3a–d). Interestingly, detailed analysis of the timing of the initial detectable signal changes revealed that the surface potential change significantly preceded the onset of the Ca^{2+} signal in the plant GLV sensory transduction system (Fig. 3e).

Revised main text: Page 7: Interestingly, changes in the leaf surface potential preceded changes in $[Ca^{2+}]_{cyt}$ upon GLV exposure (Fig. 3e), as systemic electrical

signals preceded Ca^{2+} signals upon wounding^{54, 55}. Membrane depolarization could be induced by the activation of ion channels, such as Cl^- -permeable, ROS-sensitive, or ligand-gated channels, followed by $[\text{Ca}^{2+}]_{\text{cyt}}$ increases via Ca^{2+} influx and efflux through plasma- and endo-membranes, respectively^{54, 55}.

-L128. I think, aside from Figure 4a, it is important to show box plots where the comparison of the Ca^{2+} peaks observed in response to the different concentrations is reported. Moreover, I would also like to see a comparison of the times at which the Ca^{2+} peaks are observed. Surely, a proper statistical analysis must be presented.

We greatly appreciate these valuable suggestions. We presented additional data comparing the Ca^{2+} peaks observed in response to different concentrations of Z-3-HAL. Unfortunately, when analyzing the times at which the Ca^{2+} peaks appeared, no concentration dependency was noted. This is because the Ca^{2+} changes did not peak within 600 s under our experimental conditions, and the maximal value for most samples was detected at 600 s. Instead, we presented additional data specifically comparing the time required to reach the maximal Ca^{2+} changes following exposure to different concentrations of Z-3-HAL. By analyzing the time required to reach 50% of the mean maximal value, we observed a trend of concentration dependency.

Revised main text: Page 21: (b) Comparison of the maximal $[\text{Ca}^{2+}]_{\text{cyt}}$ changes detected in receiver *Arabidopsis* upon exposure to each concentration of Z-3-HAL. Error bars, mean \pm SE. Different letters denote significant differences based on one-way ANOVA followed by Tukey's honestly significant difference post hoc test ($P < 0.05$).

Revised supplementary text: Page 4: Supplementary Fig. 4. Comparison of the time required to reach the mean value of maximal $[\text{Ca}^{2+}]_{\text{cyt}}$ changes detected in receiver *Arabidopsis* upon exposure to different concentrations of Z-3-HAL.

Although the time required to reach the mean value of maximal $[\text{Ca}^{2+}]_{\text{cyt}}$ changes upon exposure to different concentrations of Z-3-HAL did not exhibit concentration dependency (**a**), a trend of concentration dependency was observed in the analysis of the half-maximal response (defined as the time required to reach 50% of the mean maximal value) (**b**). A value of 600 s was assigned to represent those samples that did not reach the mean value. Error bars, mean \pm SE. $N > 7$. Different letters denote statistically significant differences based on one-way ANOVA followed by Tukey's honestly significant difference post hoc test ($P < 0.05$). ns, not significant.

-L128. What concentration of (Z)-3-Hexenal was used? This information is reported only in the M&M and it seems they used 0.25 M. This is strange because in Figure 4d treatment with 0.25 M induces a transient where the $\text{DF}/\text{F0}$ reaches 1.0, where the same concentration in Figure 4a triggers only a $\text{DF}/\text{F0}$ of 0.3. Moreover, can the authors explain which was the

rationale for using 0.25 M among the other concentrations tested? Theoretically, also for the investigation of systemic responses, a dose-dependent analysis would be useful.

We apologize for our insufficient explanation regarding the experimental conditions. As you mentioned, 0.25 M Z-3-HAL was used in the small cuvette experiment (Fig. 4c) because the viability of the plant material in this small cuvette was affected by 3.0 M Z-3-HAL, which was the concentration used in the open-air experiments. Instead, we redesigned the experimental setup, as described schematically in Fig. 4c, in which a new square plastic dish was processed to spatially segregate and expose only L1 to 3.0 M Z-3-HAL. In addition, we included data comparing the maximum $\Delta F/F_0$ values for each ROI. The manuscript was revised to reflect these points.

Revised main text: Page 10: To selectively treat L1 with Z-3-HAL, a hole was created on the side of a square plastic dish with a cutter, and L1 was spatially segregated from other parts of the plant by inserting it into the hole, as described in Fig. 4c. Then, the hole was filled with 0.2% agarose gel, and Ca^{2+} imaging was performed.

Revised main text: Page 21: (d) The time-course changes in $[\text{Ca}^{2+}]_{\text{cyt}}$ (yellow arrowheads) in L1 and L3 leaves after exposing L1 to Z-3-HAL. The dashed outline indicates the position of the square dish. Scale bar, 5 mm.

(f) Comparison of the maximal $[\text{Ca}^{2+}]_{\text{cyt}}$ changes detected in L1 (ROI 1) and L3 (ROI 2) of receiver *Arabidopsis* upon exposure to 3.0 M Z-3-HAL. Error bars, mean \pm SE. N = 6. An asterisk denotes statistically significant differences based on Student's *t*-test (*, $P < 0.05$).

-L144. Also, in this case, I would suggest the authors to better analyze and discuss the data. I would not only show Ca^{2+} traces, but I would compare the times at which the Ca^{2+} increase starts in the different sensor lines. Also comparing the maximum Ca^{2+} peaks for the different regions of the different lines would be useful. Showing the data in this way would help to better discuss the results. Also, the choice of the selected images at different time points is weird. I would select the same time points (e.g. 0, 150, 300, 600, and 900 s) for all lines.

As you recommended, we added data comparing the timing of the onset of Ca^{2+} signals in the tip region of leaf 1 for each transgenic line. Ca^{2+} signals were detected in guard and mesophyll cells before they appeared in epidermal cells, and the difference in timing was statistically significant. We sincerely appreciate your valuable suggestions regarding the maximal Ca^{2+} peaks. Unfortunately, we did not observe any significant differences among the transgenic lines because the Ca^{2+} changes did not peak within 600 s, and the maximal value of most samples was detected at 600 s. Therefore, we would prefer not to incorporate this information to the current manuscript. As you requested, we have revised our manuscript to

include additional images comparing the Ca^{2+} signals at the same time point across transgenic lines.

Revised main text: Page 23: (f) Enlarged graph displaying the onset of $[\text{Ca}^{2+}]_{\text{cyt}}$ increases within 160 s in the tip region of each transgenic line. (g) Comparison of the time points at which the increase in the initial Ca^{2+} signal was detected in the tip region of L1 of *pGCl::GCaMP3* (G), *pRBCS1A::GCaMP3* (R), *pSULTR2;2::GCaMP3* (S), and *pATML1::GCaMP3* (A), following Z-3-HAL exposure. The increase in signal used to calculate velocity was defined as an increase to above 2 SD of the pre-stimulation levels. Error bars, mean \pm SE. $N > 5$. Different letters denote statistically significant differences based on one-way ANOVA followed by Tukey's honestly significant difference post hoc test ($P < 0.05$).

-L147. I have the same comments made for Figure 5. A better analysis of the data is needed. As a minor comment, there is a need to indicate that the Overlay is between the GFP and bright field channels.

We agree with your remarks. We have provided the overlay image and details in the legend. Based on your previous comment, we included additional data comparing the timing of the onset of Ca^{2+} signals at the cellular level for each transgenic line (Fig. 6h).

Revised main text: Page 25: (g) Quantification of the $[\text{Ca}^{2+}]_{\text{cyt}}$ levels in the L1 of each transgenic line. Error bars, mean \pm SE. $N = 5$. (h) Comparison of the time points at which the increase in the Ca^{2+} signal was detected in *pGCl::GCaMP3* (G), *pRBCS1A::GCaMP3* (R), and *pATML1::GCaMP3* (A) following Z-3-HAL exposure. The increase in signal used to calculate velocity was defined as an increase to above 2 SD of the pre-stimulation levels. Error bars, mean \pm SE. $N = 5$. Different letters denote statistically significant differences based on one-way ANOVA followed by Tukey's honestly significant difference post hoc test ($P < 0.05$).

Revised main text: Page 25: The merged GFP signal and bright field images were presented as an overlay.

-L157. I would introduce at the beginning of the paragraph the rationale behind the use of ABA and mutants impaired in ABA-induced stomatal closure. I would also highlight in the text that for this experiment, the GCaMP3 was under the control of the CaMV35S promoter. As for the other figures, a box plot for the comparison of the Ca^{2+} peaks should be presented.

We appreciate your valuable suggestion. We have revised the manuscript to emphasize the strategy of using ABA and mutants with impaired ABA-induced stomatal closure to investigate the role of stomata in rapid responses to GLVs in *Arabidopsis*. Additionally, we have revised the manuscript to emphasize the use of *Arabidopsis* expressing GCaMP3 driven under 35S promoter. Furthermore, we have added data comparing the

maximal Ca^{2+} changes for leaves treated with either ABA or Mock and observed a decrease in the maximal Ca^{2+} changes in WT leaves pretreated with ABA than in mutant leaves and WT leaves treated with Mock.

Revised main text: Page 5: To examine the function of stomata in rapid GLV sensory transduction, we used the phytohormone abscisic acid (ABA) to induce stomatal closure³⁷ as well as stomatal mutants exhibiting a constitutive stomata opening phenotype^{38,39}. Pretreatment of *Arabidopsis* WT leaves with ABA resulted in stomatal closure (Supplementary Fig. 7), and the Z-3-HAL-induced increase in $[\text{Ca}^{2+}]_{\text{cyt}}$ was delayed compared with that in leaves pretreated with a stomatal opening buffer solution (Mock, Fig. 7). Loss-of-function mutants of *SLOW ANION CHANNEL-ASSOCIATED 1* (*slac1*) and *OPEN STOMATA 1* (*ost1*) genes exhibit impaired stomatal closure in the presence of ABA^{38, 39} (Supplementary Fig. 7). Z-3-HAL-induced $[\text{Ca}^{2+}]_{\text{cyt}}$ signatures in ABA-treated *slac1-2* and *ost1-3* mutants expressing GCaMP3 driven by the 35S promoter were similar to those in water-treated leaves (Fig. 7).

Revised main text: Page 27: (c) Comparison of the maximal $[\text{Ca}^{2+}]_{\text{cyt}}$ changes detected in the detached leaves of WT (W), *slac1-2* (S), and *ost1-3* (O). Error bars, mean \pm SE. $N > 5$. Different letters denote statistically significant differences based on one-way ANOVA followed by Tukey's honestly significant difference post hoc test ($P < 0.05$).

-L326. 16 hrs treatment with 50 mM of La^{3+} or EGTA is a very strong condition, potentially harmful for plants. Maybe, lower concentrations should be tested.

Thank you for indicating these important points. We investigated lower concentrations of La^{3+} and EGTA (<50 mM), but their inhibitory effects were variable (not stable), indicating that 50 mM was the lowest concentration at which Ca^{2+} signals were completely blocked in our experiments. Because these chemicals are applied only to the roots, a high concentration (50 mM) of La^{3+} or EGTA is required for transport/diffusion throughout the plant body. Therefore, we conducted new experiments evaluating the viability of plants treated with 50 mM LaCl_3 or EGTA using the washout assay, a commonly used method in physiological/pharmacological research. After washing the plants with a liquid medium lacking the pharmacological reagents, we observed significant recovery of Ca^{2+} signals in leaves pretreated with LaCl_3 or EGTA. This reversible result indicated that these chemicals are unlikely to kill plants and that their potential harmful effects are negligible.

Revised main text: Page 4: Furthermore, we conducted the washout assay to evaluate the reversibility of the Ca^{2+} signals in *Arabidopsis*, allowing the assessment of any potential negative effects of these pharmacological reagents on plant cell viability. After an additional incubation period of 24 h with a liquid medium lacking LaCl_3 and

EGTA, we observed approximately 65%–95% recovery of the Ca^{2+} signal at 500 s compared with that in Mock-pretreated *Arabidopsis* (Supplementary Fig. 3a, c). Therefore, the GLV response in plants pretreated with the reagents is reversible, pharmacologically suggesting that these chemicals can block Ca^{2+} signals without drastically affecting the plant cells themselves. Altogether, these results indicate that $[\text{Ca}^{2+}]_{\text{cyt}}$ increases are required for the induction of transcriptional changes related to defense responses in plants.

Revised main text: Page 11: For the washout assay, *Arabidopsis* seedlings that had been treated with 50 mM LaCl_3 or EGTA were uprooted from MS medium and washed 5 times with liquid MS medium devoid of these inhibitors to remove any residual inhibitors. Subsequently, the seedlings were transferred to an inhibitor-free MS agar plate and incubated for 24 h before conducting Ca^{2+} imaging.

Revised supplementary text: Page 3: After incubation of these seedlings with a medium without the pharmacological reagents for 24 h, the reversibility of Ca^{2+} signals was observed (right). Dashed white lines indicate the position of the plastic tube.

-L382. *It is important to report in the Figure legend that Arabidopsis ACT8 was used as an internal reference for qRT-PCR standardization.*

We appreciate this suggestion. We revised the legend accordingly.

Revised supplementary text: Page 3: *ACT8 was used as an internal reference for standardization.*

Supplementary Figure 3. In this case, the response to (E)-2-Hexenal is stronger than the (Z)-3-Hexenal a result that is opposite to what is shown in Figure 3. The authors need to comment on this.

We sincerely appreciate this critical comment. In the pharmacological experiments, younger (8 days old) plants were used because the chemical agents were introduced throughout the plant body and obvious inhibitory effects were observed. However, 2-week-old plants were used for Ca^{2+} imaging, electrical measurements, and qPCR analyses. GLV-induced *HSP90.1* expression might be age-dependent. However, differences in the *HSP90.1* induction rate did not affect our hypothesis, as both Z-3-HAL and E-2-HAL evidently induced *HSP90.1* in *Arabidopsis*; this effect was inhibited by LaCl_3 and EGTA. The experimental conditions have been described in the revised manuscript.

Revised main text: Page 12: To extract total RNA, L1 and aboveground parts were harvested from 2-week-old *Arabidopsis* plants and 8-day-old *Arabidopsis* seedlings,

respectively (Fig. 3f and Supplementary Fig. 3b). The harvested tissues were immediately frozen using liquid nitrogen.

Supplementary Figure 4. I would suggest moving this panel to the main Figure 3 and performing a statistical analysis by comparing the times at which they observe the cytosolic Ca²⁺ increase in the different lines.

We appreciate your valuable suggestion. As requested, we have moved Supplementary Fig. 4 to the main text as Fig. 5f. Additionally, we performed a statistical analysis comparing the onset times of Ca²⁺ changes in the transgenic lines. This analysis revealed that Ca²⁺ signals appeared earlier in pGC1 and pRBCS1A than in pATML1 (Fig. 5g).

Revised main text: Page 23: (f) Enlarged graph displaying the onset of [Ca²⁺]_{cyt} increases within 160 s in the tip region of each transgenic line. **(g)** Comparison of the time points at which the increase in the initial Ca²⁺ signal was detected in the tip region of L1 of *pGC1::GCaMP3* (G), *pRBCS1A::GCaMP3* (R), *pSULTR2;2::GCaMP3* (S), and *pATML1::GCaMP3* (A), following Z-3-HAL exposure. The increase in signal used to calculate velocity was defined as an increase to above 2 SD of the pre-stimulation levels. Error bars, mean ± SE. N > 5. Different letters denote statistically significant differences based on one-way ANOVA followed by Tukey's honestly significant difference post hoc test ($P < 0.05$).

Reviewer #2:

In this manuscript Aratani et al. report their investigation of an interesting aspect of plant-to-plant communication, which is the triggering of defense reactions by volatile organic compounds (VOCs) in plants. Specifically, the authors used wide-field real-time imaging of GCaMP3-expressing plants to clarify the role of Ca²⁺ signaling and its spatio-temporal dynamics in plant-to-plant communication. They report that VOCs emitted from homogenized leaves induce local Ca²⁺ signals in exposed plants. Moreover, they confirm that (Z)-3-Hexenal and (E)-2-Hexenal induce Ca²⁺ signals and membrane depolarization. Finally, by using plant lines that express Ca²⁺ reporter proteins under control of cell-type specific promoters, the authors provide evidence that stomata indeed provide entrance points for VOCs that elicit these Ca²⁺ signals. Overall, the manuscript is well written, and the experiments are principally well executed and documented. I personally like the movies very much. I have the following major comments:

-The authors use crushed leaf homogenates to induce Ca²⁺ signals in the exposed plants. This technical setup falls short to really establish that such a plant-to-plant communication (that evokes Ca²⁺ signals in the receiving plant) indeed occurs in nature. To uphold this claim, it would require to use an experimental setup as it has been reported for example in ref. 26 (Zebelo et al., 2012; Figure 1). Here, an airflow was directed over a plant, on which caterpillars were feeding and then subsequently directed to the plant that was analyzed.

We sincerely appreciate your valuable and thoughtful comments. Following these suggestions, we conducted additional experiments to investigate the possibility of Ca^{2+} increases in plants under realistic conditions upon exposure to VOCs emitted by damaged plants. To newly establish this experimental setup, we referred to the method described in a study by Dr. Maffei's team (Zebelo et al., 2012; Fig. 1a). When receiver *Arabidopsis* was exposed to VOCs released from tomato and *Arabidopsis* leaves consumed by the herbivore *S. litura*, we visualized obvious Ca^{2+} increases in *Arabidopsis* in real time (Fig. 1b). These results support our hypothesis that Ca^{2+} signals occur in plants under natural conditions through exposure to VOCs released from mechanically damaged and herbivore-damaged plants. We have added these data, movies, and descriptions to the revised text.

Revised main text: Page 3: First, we monitored $[\text{Ca}^{2+}]_{\text{cyt}}$ increases in *Arabidopsis* (receiver) following exposure to VOCs emitted from *Arabidopsis* plants (source of VOCs) fed on by the common cutworm (*Spodoptera litura*; Fig. 1a, b and Supplementary Video 1). This Ca^{2+} signal was rapidly transmitted to all parts of the plant within 20 min (Fig. 1c). Similar results were obtained when receiver *Arabidopsis* was exposed to VOCs released by tomato leaves (source of VOCs) consumed by *S. litura* (Fig. 1b, c and Supplementary Video 2).

Revised main text: Page 15: Fig. 1. Exposure to VOCs emitted by homogenized leaves and leaves consumed by the herbivore *S. litura* induces $[\text{Ca}^{2+}]_{\text{cyt}}$ increases in receiver *Arabidopsis* leaves.

(a) The experimental setup for Ca^{2+} imaging in *Arabidopsis* (receiver) upon exposure to VOCs emitted by leaves consumed by *S. litura* larvae is schematically illustrated. Prior to the experiment, the receiver *Arabidopsis* in a plastic dish was acclimated by directing airflow from an empty plastic bottle for 10 min, allowing its adaptation to the experimental conditions. Subsequently, receiver *Arabidopsis* was exposed to VOCs emitted from a plastic bottle containing *S. litura* larvae and either *Arabidopsis* or tomato leaves (source of VOCs) by connecting the bottle and manipulating the valve. The black arrow indicates the direction of airflow. (b) Changes in $[\text{Ca}^{2+}]_{\text{cyt}}$ (yellow arrowheads) in *Arabidopsis* expressing GCaMP3 in response to VOCs released from *Arabidopsis* (upper) and tomato (*Solanum lycopersicum*) leaves (below) consumed by *S. litura* larvae. White dashed lines indicate the position of the tip of the tube from where the airflow emerges. Scale bar, 5 mm. (c) Quantification of $[\text{Ca}^{2+}]_{\text{cyt}}$ signatures in leaf 1 (L1). Error bars, mean \pm SE. $N > 3$.

- The result that VOC exposure triggers Ca^{2+} signals in exposed plants is not really novel and has principally been reported by Zebelo et al., 2012 and (with aequorin plants) by Asai et al., 2009. Admittedly, the data that are provided here are of much higher quality and resolution. Also, (Z)-3-Hexenal and (E)-2-Hexenal have been intensively studied in the past and have already previously been reported to be the compounds that trigger Ca^{2+} signals and membrane depolarization (Zebelo et al., 2012).

- The authors detail here that guard cells represent the entry points of VOCs and that consequently, VOC-triggered Ca^{2+} signals start around these leaf valves. These are now very convincing data. Nevertheless, with more indirect evidence a role of guard cell conductance for VOC uptake has been indicated before.

We greatly appreciate your positive evaluation of our study. Although it is true that previous pioneering studies by Dr. Maffei's and Dr. Takabayashi's teams have detected Ca^{2+} signals in response to VOCs using a fluorescent dye (calcium orange) or Ca^{2+} -sensitive luminescent protein (apoaquorin), we made significant improvements in various aspects of this research field. We presented real-time visualization of Ca^{2+} dynamics induced by VOCs with significantly higher resolution using intact *Arabidopsis*, allowing the assessment of Ca^{2+} signaling in a more comprehensive manner. Furthermore, our imaging and electrophysiological methods in combination with pharmacological and molecular genetic techniques revealed spatial and temporal information regarding the pathway of VOC perception in plants at the cellular level, providing evidence that stomata act as entry points in VOC sensory signal transduction in *Arabidopsis*. We have revised the manuscript to emphasize the novelty and importance of our findings.

Revised main text: Page 8: The wide-field real-time imaging approach used in this study provided novel physiological insights with significantly higher resolution using intact *Arabidopsis*, revealing the details of Ca^{2+} signals in response to GLVs in a more comprehensive manner. This methodology allowed the assessment of spatial and temporal aspects of GLV perception pathways at the cellular level. Additionally, by integrating real-time imaging with other techniques, we achieved a deeper understanding of the comprehensive orchestration of GLV responses, including Ca^{2+} signals and other signaling mechanisms such as electrical signals.

Reviewer #3:

The response of plants to VOC is known since long time and the effects of green leaf volatiles on calcium variations have already been demonstrated in other plant species. Therefore, the results here presented do not represent a novelty and confirm previous studies. The only further information given here is the sequential activity of calcium release and the possible determination of guard cells as the first VOC sensing cells.

The work is of significance in the general response of plants to airborne VOCs because it provides further evidence of the involvement of calcium in plant-insect and plant-environment interactions. However, with regards to the specific literature it lacks one important information, the quantitative and qualitative determination of VOC emitted by the homogenized material here used. This is also affecting all results obtained with the presumed molecules and the unrealistic concentrations used. Without the chemical characterization of the emitted VOCs the data here presented lack of realism, because hundred of VOCs are emitted by plants. In fact, by comparing the presented results with those of the literature (see for instance <https://doi.org/10.1074/jbc.ra118.005843>) the concentrations used are about 1000 times as expected. Nevertheless, for many other technical aspects the work is original.

Due to the above criticism (above all the lack of any chemical characterization and quantification) many of the conclusions are unsustainable and before publication additional experiments are needed in order to assess: 1) the exact chemical composition and the quantification of VOCs released from homogenized tissues (both Arabidopsis and Tomato). 2) The quantification of how much of these VOCs are volatilized in the experimental area (very often the authors write “in close proximity”, this description means nothing in terms of repeatability of the experiment). The methodology related to the detection of Calcium variations is sound and, specifically to the calcium signaling area, the work meets the expected standards in the plant science field. The methods are appropriate.

We sincerely appreciate these valuable comments and suggestions. We acknowledge that Dr. Maffei's team conducted pioneering research on tomatoes approximately 10 years ago (Zebelo et al., 2012) and provided new insights into plant physiology. Their study revealed the involvement of Ca²⁺ variations in plant VOC perception systems. However, we believe that our research has made some significant advances in this field, particularly in terms of improving the quality of data using new imaging technology and enhancing our understanding of the mechanisms by which plants perceive VOCs at both tissue and cellular levels.

A key contribution of our study is the development of an advanced imaging method in combination with electrophysiological, pharmacological, and molecular genetic techniques, enabling real-time, high-resolution, cell type-specific Ca²⁺ imaging in intact *Arabidopsis* plants upon GLV exposure. This new technology provides spatial and temporal information regarding plant VOC sensory signaling and unveils the comprehensive orchestration of GLV responses via stomata, including Ca²⁺ and electrical signals. Therefore, we believe that this multidimensional approach provides important steps in understanding when, where, and how plants sense VOCs in plant-to-plant communications. We have revised the manuscript to emphasize the novel contribution and significance of this research in the advancement of this research field.

We also appreciate your suggestions to quantify the composition and amount of VOCs in our experimental conditions. Following your suggestion, we recruited Dr. Kenji Matsui (Yamaguchi University, Japan), a plant biologist working on GLVs, and analyzed the atmospheric VOC content emitted from homogenized leaf tissues in tomato and *Arabidopsis* WT and *hpl1* mutants and VOC-containing solutions. By quantifying and characterizing the atmospheric VOC content under our experimental conditions, we successfully detected the rapid and substantial emission of Z-3-HAL and E-2-HAL from homogenized tomato and *Arabidopsis* samples. Furthermore, based on the results obtained from *hpl1* mutants and WT plants, we calculated the GLV concentration necessary to induce Ca²⁺ signaling in *Arabidopsis*. Moreover, we specifically quantified the amount of Z-3-HAL released from the solution at a distance of 5 mm, which corresponds to the position of the leaves under our experimental conditions. By determining the amount of Z-3-HAL to which *Arabidopsis* is exposed within a short period, we could estimate the Z-3-HAL content required to induce Ca²⁺ signaling in *Arabidopsis*. Based on these findings, we discussed the sensitivity of plants to GLVs, particularly with regard to Ca²⁺ signaling. We also addressed the possibility that the observed

Ca²⁺ signals could occur under realistic natural conditions. Details of these revisions are provided in a point-by-point manner.

We again sincerely appreciate your valuable comments and suggestions, which have helped improve the clarity and importance of our findings.

Other issues:

-the authors report experiments with beta-pinene (Line 92) but no results are reported for this monoterpene

We apologize for this oversight. We have included data regarding beta-pinene and revised Fig. 2f accordingly.

-The paragraph of lines 111-115 is the repetition of previous results and should be deleted since it does not represent novelty.

In a previous pioneering study, Dr. Maffei's team (Zebelo et al., 2012) reported VOC-induced Ca²⁺ and electrical signals in tomatoes. However, we believe that our study provides new physiological insights into VOC sensory Ca²⁺ and electrical signaling in plants. Compared with the previous studies, our study presents several novel findings:

1. We developed a unique simultaneous recording system to analyze both Ca²⁺ and electrical dynamics in plants, providing a real-time spatial and temporal correlation between Ca²⁺ and electrical signals induced by GLVs in plants.
2. This simultaneous recording system enabled the precise analysis of the timing of the onset of Ca²⁺ and electrical signals and revealed that electrical signals precede Ca²⁺ signals.
3. We discovered that GLVs induce Ca²⁺ and electrical signaling in intact *Arabidopsis* plants.
4. These results are crucial for future experiments investigating the roles of stomatal regulation in VOC-induced Ca²⁺ and electrical signaling in plants (e.g., Supplementary Fig. 8).

We have revised the manuscript to emphasize the novelty and significance of our findings in this field.

Revised main text: Page 4: Z-3-HAL and E-2-HAL exposure in *Arabidopsis* leaves resulted in rapid changes in the leaf surface potential, which is spatiotemporally coupled with changes in [Ca²⁺]_{cyt} (Fig. 3a–d). Interestingly, detailed analysis of the timing of the initial detectable signal changes revealed that the surface potential change significantly preceded the onset of the Ca²⁺ signal in the plant GLV sensory transduction system (Fig. 3e).

Revised main text: Page 7: Interestingly, changes in the leaf surface potential preceded changes in [Ca²⁺]_{cyt} upon GLV exposure (Fig. 3e), as systemic electrical signals preceded Ca²⁺ signals upon wounding^{54, 55}. Membrane depolarization could be

induced by the activation of ion channels, such as Cl^- -permeable, ROS-sensitive, or ligand-gated channels, followed by $[\text{Ca}^{2+}]_{\text{cyt}}$ increases via Ca^{2+} influx and efflux through plasma- and endo-membranes, respectively^{54, 55}.

-lines 122-123: this is already well known from the literature

We thank the reviewer for this valuable comment. Indeed, the relationship between Ca^{2+} signaling and transcriptional accumulation related to defense responses has been investigated in various plants. However, pharmacological approaches have not been used to determine (1) whether Ca^{2+} channel activity and apoplastic Ca^{2+} are involved in $[\text{Ca}^{2+}]_{\text{cyt}}$ increases in response to GLVs and (2) whether the $[\text{Ca}^{2+}]_{\text{cyt}}$ increases are required for the transcriptional changes. We have revised the manuscript to highlight the novelty of our findings.

Revised main text: Page 7: In this study, oxidative stress- and heat-responsive genes were upregulated upon GLV exposure in a Ca^{2+} -dependent manner (Fig. 3f and Supplementary Fig. 3b), consistent with the previous prediction that the expression of oxidative stress- and heat-responsive genes is mediated by $[\text{Ca}^{2+}]_{\text{cyt}}$ increases^{48, 49}. Additionally, GLV-exposed *Arabidopsis* seedlings exhibited enhanced heat tolerance²¹, suggesting that GLV exposure contributes to several aspects of stress tolerance. Although the detailed molecular mechanism underlying abiotic stress-related signaling activation upon GLV exposure remains unelucidated, Ca^{2+} signals act as key components mediating the transcriptional accumulation of abiotic stress-related genes in response to GLVs.

-Line 133 Supplementary video 5 shows the opposite of what reported in this sentence

We apologize for the confusion in our previous manuscript. We would like to provide a more detailed explanation of our experimental objectives and the results presented in Supplementary Video 5 (new Supplementary Video 7 in the revised manuscript). The primary aim of the experiment was to treat only leaf 1 with Z-3-HAL. In this regard, we carefully designed an experimental setup that allowed the spatial segregation of leaf 1 from the plant body. Upon analyzing the results, we observed significant Ca^{2+} signals occurring exclusively in leaf 1. As quantitatively analyzed in Fig. 4f, we could not detect any signal transmission to distal leaves from leaf 1, indicating a localized Ca^{2+} response within leaf 1 exposed to GLVs.

-Line 173. The authors have measured differences in Vm I'm not sure they can claim they showed electrical signals.

We appreciate this important comment. Surface potential changes have been extensively investigated in plants (Mousavi et al., Nature 2013). Based on numerous electrophysiological studies, surface potential changes reflect changes in membrane potential, which was confirmed by intracellular recordings (Salvador-Recatalà et al., New Phytologist

2014) and practically termed electrical signals (Farmer et al., New Phytologist 2020). Therefore, we believe it is reasonable to use the term “electrical signals” to refer to surface potential changes.

-Lines 174-175. The authors did not provide a full genome response to GLVs, therefore based only on 4 genes analyzed they cannot claim “diverse Ca²⁺-dependent defense responses”

We thank the reviewer for these remarks. We have revised the manuscript to replace the phrase “diverse Ca²⁺-dependent defense responses” with “downstream signaling pathway including defense-related transcriptional accumulation.”

Revised main text: Page 6: These data indicate that GLVs function as airborne signaling molecules that activate downstream signaling pathways including defense-related transcriptional accumulation in *Arabidopsis* leaves.

-The sentence of line 177-178 contradicts what is reported in lines 182-187. The authors did not know the true composition of the emitted VOCs therefore they do not know whether other molecules are present. Moreover, the generalization to aldehyde moiety is only supported by unrealistic concentrations, while it is known that plant respond differently to low and very low concentrations of VOCs. Indeed there are several mono and sesquiterpene aldehydes and these are known to induce defense in plants.

This is an excellent point. We have determined the composition and amount of VOCs emitted instantly from homogenized tomatoes and *Arabidopsis* using GC–MS. Our analysis revealed the significant and rapid emission of two key components (*Z*-3-HAL and *E*-2-HAL); however, C6 alcohol (*Z*-3-HOL) and acetate (*Z*-3-HAC) were not detected. Importantly, we observed a positive correlation between the emitted amounts of these C6 aldehydes and induction of Ca²⁺ signals in receiver *Arabidopsis*. These results align with our hypothesis that only *Z*-3-HAL and *E*-2-HAL have the ability to induce Ca²⁺ signals in *Arabidopsis*, suggesting the presence of a putative GLV perception system that specifically recognizes structural differences.

Based on the results of GC–MS analysis, the *Z*-3-HAL concentration emitted from 3 M solution fits in the range of that emitted from homogenized *Arabidopsis* leaf tissues. Furthermore, approximately 148.8 (31.5 mm × 31.5 mm = 992.3 mm²) and 662.1 (66.4 mm × 66.4 mm = 4409.0 mm²) mg of leaves need to be injured in tomatoes and *Arabidopsis*, respectively, to induce detectable Ca²⁺ signals. These findings and estimations support the notion that although high concentrations of *Z*-3-HAL solutions are required to consistently elicit stable [Ca²⁺]_{cyt} increases in *Arabidopsis*, the Ca²⁺ responses remained in the range of those in natural conditions.

Revised main text: Page 4: Using gas chromatography mass spectrometry (GC–MS) and MonoTrap RGPS TD, which is a high-quality adsorbent, we further analyzed the VOC components rapidly produced by homogenized tomato and *Arabidopsis* leaves.

Tomato leaves emitted significantly higher amounts of Z-3-HAL and E-2-HAL than wild-type (WT) *Arabidopsis* leaves upon homogenization (Supplementary Fig. 2). Moreover, the levels of these compounds emitted by homogenized *hpl1* leaves were remarkably low (<25% of that emitted by WT leaves; Supplementary Fig. 2), consistent with the results of Ca²⁺ signature levels.

Revised main text: Page 8: Approximately 3.9 nmol Z-3-HAL was adsorbed by adsorbents from a 0.03 M Z-3-HAL solution within 30 s (Supplementary Fig. 5). Considering the volume of the adsorbents (58 μ L), it can be estimated that the local concentration of Z-3-HAL at a distance of 5 mm from the 3 M solution, which consistently induced stable [Ca²⁺]_{cyt} increases, was 6.7 mM. Similarly, 381 nmol Z-3-HAL was detected within 10 min of exposure to 10 g homogenized *Arabidopsis* leaf tissues (Supplementary Fig. 2), indicating that receiver *Arabidopsis* is exposed to approximately 6.6 mM Z-3-HAL in 30 s, which fits in the range of the concentration emitted from the 3 M Z-3-HAL solution. Although the homogenization of *Arabidopsis* leaf tissues might be unrealistic in nature, these estimations indicate that the concentration of Z-3-HAL solution corresponds to that emitted by plants and that the experimental conditions used in this study are relevant to the potential exposure or emission of Z-3-HAL in a natural context.

Revised main text: Page 8: Based on the calculation methodology described in a previous study³¹, a single injury (7.5 μ g, 0.05 mm²) produces 12.9 and 2.9 pmol Z-3-HAL in tomatoes and *Arabidopsis*, respectively (for detail, see Supplementary Fig. 2 and Methods). If these estimations are accurate, then the Z-3-HAL emitted following a single injury are unlikely to induce detectable Ca²⁺ signals. To induce the release of 12.8 nmol Z-3-HAL (the accumulation of 4.2 nmol Z-3-HAL in *Arabidopsis* leaves) and Ca²⁺ signals under our experimental conditions, approximately 148.8 (31.5 mm \times 31.5 mm = 992.3 mm²) and 662.1 (66.4 mm \times 66.4 mm = 4409.0 mm²) mg of leaves need to be injured in tomatoes and *Arabidopsis*, respectively. This size of agricultural damage might be realistic in nature⁵⁷.

Revised main text: Page 13: To estimate Z-3-HAL production from a single injury, we followed a previously described method³¹. This estimation was based on the detection of Z-3-HAL that elicits [Ca²⁺]_{cyt} increases in *Arabidopsis* following leaf homogenization as described in Supplementary Fig. 2. The total amounts of Z-3-HAL produced by homogenized *Arabidopsis* and tomato leaves were quantified as 380.9 and 1718.7 nmol/gFW, respectively. These values represent the maximum capacity of Z-3-HAL synthesis by the leaves. Based on previous research³¹, the average weight of a leaf was defined as 150 μ g/mm². Considering the estimated area of a single injury to be 0.05 mm² as previously described³¹, we calculated that a single wound would result in the production of approximately 2.9 and 12.9 pmol of Z-3-HAL for *Arabidopsis* and tomatoes, respectively.

-Lines 206-207. The authors did not provide any evidence that stomata “activate plant defense responses”, they are probably involved but surely not the only one involved. With the presented data this sentence is pure speculation.

We apologize for the oversight. We have revised the manuscript to soften the wording.

Revised main text: Page 7: Taken together with the previous findings, it is possible that stomata serve as a plant gateway mediating rapid VOC entry into interspaces in tissues, which could induce changes in the physiological properties of plants such as $[Ca^{2+}]_{cyt}$ increases.

-Lines 212-215. The authors did not provide any evidence of an oxidative stress or heat stress related to VOC perception. They simply tested a couple of genes and speculated over the full process of responses to ROS and heat.

We apologize for the oversight. In our study, we investigated the induction of marker genes in response to GLV exposure in *Arabidopsis*, highlighting the relationship between Ca^{2+} signaling and transcriptional stress responses. Although we acknowledge that Yamauchi et al. demonstrated the induction of thermotolerance in plants exposed to GLVs and comprehensive upregulation of heat tolerance- and oxidative stress-related genes, we believe that additional investigation is necessary to establish a direct link between GLV-induced Ca^{2+} signals and heat tolerance or oxidative stress responses. Consequently, we have revised the manuscript to address the need for future studies to explore the potential involvement of GLV-induced Ca^{2+} signals in heat tolerance and oxidative stress responses.

Revised main text: Page 7: Additionally, GLV-exposed *Arabidopsis* seedlings exhibited enhanced heat tolerance²¹, suggesting that GLV exposure contributes to several aspects of stress tolerance. Although the detailed molecular mechanism underlying abiotic stress-related signaling activation upon GLV exposure remains unelucidated, Ca^{2+} signals act as key components mediating the transcriptional accumulation of abiotic stress-related genes in response to GLVs.

-Line 216. The authors evidently ignore that Calcium signals are known to be key components in VOC signaling since tens of years.

We apologize for this error. In this paragraph, we wanted to discuss various GLV-related abiotic stress responses, such as heat and light responses. We have cited the appropriate previous articles and revised the sentence accordingly.

Revised main text: Page 7: Although the detailed molecular mechanism underlying abiotic stress-related signaling activation upon GLV exposure remains unelucidated, Ca^{2+} signals act as key components mediating the transcriptional accumulation of abiotic stress-related genes in response to GLVs.

-Lines 234-237. At last the authors realized that without the careful quantitative and qualitative characterization of the VOC emitted any consideration (especially when brought by unrealistic concentrations) remains purely speculative. Based on the above consideration, the manuscript cannot be accepted for publication.

We appreciate these valuable suggestions. As previously mentioned, we conducted quantitative and qualitative analyses of GLVs emitted from homogenized leaf tissues and GLV-containing solutions. These analyses played a crucial role in the estimation of the amount of atmospheric GLVs necessary for the induction of Ca^{2+} signals in intact *Arabidopsis* plants. By quantifying the amount of GLVs emitted from solution and considering the amount emitted from homogenized leaves, we evaluated the potential occurrence of similar GLV concentrations under natural conditions. This assessment provided insights into the ecological relevance of our experimental findings and enhanced our understanding of the potential biological effects of GLV-induced signaling in intact plants. We again thank the reviewer for highlighting the importance of this comparison. We have included this information in the Discussion section to emphasize the ecological implications of our study and the potential significance of our experimental conditions to natural settings.

Revised main text: Page 4: Using gas chromatography mass spectrometry (GC–MS) and MonoTrap RGPS TD, which is a high-quality adsorbent, we further analyzed the VOC components rapidly produced by homogenized tomato and *Arabidopsis* leaves. Tomato leaves emitted significantly higher amounts of Z-3-HAL and E-2-HAL than wild-type (WT) *Arabidopsis* leaves upon homogenization (Supplementary Fig. 2). Moreover, the levels of these compounds emitted by homogenized *hpl1* leaves were remarkably low (<25% of that emitted by WT leaves; Supplementary Fig. 2), consistent with the results of Ca^{2+} signature levels.

Revised main text: Page 5: We quantified the amount of Z-3-HAL released from Z-3-HAL solution under this experimental condition. Z-3-HAL was adsorbed by adsorbents placed at a distance of 5 mm from the 0.03 M solution. Z-3-HAL adsorption reached saturation within 1 min (Supplementary Fig. 5), and 3.85 nmol Z-3-HAL was adsorbed over 30 s. Based on this finding, L1 in receiver *Arabidopsis* was exposed to approximately 6.42–385 nmol Z-3-HAL over 30 s following exposure to 0.05–3 M Z-3-HAL solutions (Fig. 4a).

Revised main text: Page 8: Approximately 3.9 nmol Z-3-HAL was adsorbed by adsorbents from a 0.03 M Z-3-HAL solution within 30 s (Supplementary Fig. 5). Considering the volume of the adsorbents (58 μL), it can be estimated that the local concentration of Z-3-HAL at a distance of 5 mm from the 3 M solution, which consistently induced stable $[\text{Ca}^{2+}]_{\text{cyt}}$ increases, was 6.7 mM. Similarly, 381 nmol Z-3-HAL was detected within 10 min of exposure to 10 g homogenized *Arabidopsis* leaf tissues (Supplementary Fig. 2), indicating that receiver *Arabidopsis* is exposed to

approximately 6.6 mM Z-3-HAL in 30 s, which fits in the range of the concentration emitted from the 3 M Z-3-HAL solution. Although the homogenization of *Arabidopsis* leaf tissues might be unrealistic in nature, these estimations indicate that the concentration of Z-3-HAL solution corresponds to that emitted by plants and that the experimental conditions used in this study are relevant to the potential exposure or emission of Z-3-HAL in a natural context.

Plants possess the ability to efficiently uptake and adsorb a wide range of surrounding atmospheric VOCs and accumulate them into their tissues. For example, *Arabidopsis* can accumulate approximately 1 $\mu\text{mol/gFW}$ (fresh weight) C6 compounds when exposed to 100 nmol/cm^3 Z-3-HAL⁵⁶. Similarly, tomato can absorb a significant amount of atmospheric methacrolein, with estimates ranging from 33% to 41% of the total methacrolein content present in the air⁵⁷. The minimum amount of Z-3-HAL required to induce detectable Ca^{2+} signals was 0.1 M (Fig. 4a). Considering that approximately 12.8 nmol Z-3-HAL was adsorbed by the adsorbents within 30 s of exposure to 0.1 M Z-3-HAL, it can be assumed that *Arabidopsis* accumulates approximately 4.2 nmol Z-3-HAL when it possesses 33% of the VOC-adsorbing capacity of the adsorbent. Based on the calculation methodology described in a previous study³¹, a single injury (7.5 μg , 0.05 mm^2) produces 12.9 and 2.9 pmol Z-3-HAL in tomatoes and *Arabidopsis*, respectively (for detail, see Supplementary Fig. 2 and Methods). If these estimations are accurate, then the Z-3-HAL emitted following a single injury are unlikely to induce detectable Ca^{2+} signals. To induce the release of 12.8 nmol Z-3-HAL (the accumulation of 4.2 nmol Z-3-HAL in *Arabidopsis* leaves) and Ca^{2+} signals under our experimental conditions, approximately 148.8 (31.5 mm \times 31.5 mm = 992.3 mm^2) and 662.1 (66.4 mm \times 66.4 mm = 4409.0 mm^2) mg of leaves need to be injured in tomatoes and *Arabidopsis*, respectively. This size of agricultural damage might be realistic in nature⁵⁷.

Although it is unlikely that plants are continuously exposed to high GLV concentrations (e.g., 3 M Z-3-HAL) under natural conditions, it should be noted that GLVs do not easily diffuse because of their high molecular weight, which may lead to high local GLV concentrations around damaged plants^{50, 58}. Considering these findings, the possibility that plant cells and tissues are temporarily exposed to high GLV concentrations under specific circumstances (e.g., when receiver plants are in close proximity to disrupted plants capable of emitting numerous GLVs, including *Vigna radiata* and *Momordica charantia*⁵⁹, in response to herbivory in nature) cannot be dismissed. In fact, we detected Ca^{2+} signals in response to VOCs released from plants consumed by herbivores (Fig. 1a–c). To further clarify this phenomenon, it would be beneficial to employ advanced technologies, such as the real-time detection of atmospheric VOC concentrations, and precisely determine the VOC adsorption capacity and efficiency of plants⁶⁰.

Revised main text: Page 13: To estimate Z-3-HAL production from a single injury, we followed a previously described method³¹. This estimation was based on the detection of Z-3-HAL that elicits $[Ca^{2+}]_{cyt}$ increases in *Arabidopsis* following leaf homogenization as described in Supplementary Fig. 2. The total amounts of Z-3-HAL produced by homogenized *Arabidopsis* and tomato leaves were quantified as 380.9 and 1718.7 nmol/gFW, respectively. These values represent the maximum capacity of Z-3-HAL synthesis by the leaves. Based on previous research³¹, the average weight of a leaf was defined as 150 $\mu\text{g}/\text{mm}^2$. Considering the estimated area of a single injury to be 0.05 mm^2 as previously described³¹, we calculated that a single wound would result in the production of approximately 2.9 and 12.9 pmol of Z-3-HAL for *Arabidopsis* and tomatoes, respectively.

REVIEWERS' COMMENTS

Reviewer #1 (Remarks to the Author):

This new version of the manuscript is much more accurate in the presentation of data and importantly includes new experiments and analyses that have improved its quality.

The story is now more solid.

Having said that, I have identified a few minor things that require authors' attention.

Line 183. "stomatal" instead of "stomata".

Line 183. "...as well as stomatal mutants exhibiting a constitutive stomata opening phenotype...".

This is not really correct for SLAC1. It was shown that the *slac1* mutant is also impaired in the stomatal opening as reported in Laanemets et al. (New Phytologist (2013) 197: 88–98 doi: 10.1111/nph.12008).

Line 258. "These results suggest that in GLV responses, there is a possibility that GLVs do not activate the key elements required for long-distance signal transmission, such as GLRs, or that signals cannot reach the core in the vasculature".

Based on our direct experience, to induce long-distance signals in response to wounding you need to damage the main vein. The distance the SWP can cover depends on the magnitude of the stimulus. It is thus plausible that VOCs do not generate a local response sufficient to reach the threshold required to allow the SWP to move out of the local leaf.

Reviewer #2 (Remarks to the Author):

The newly submitted manuscript has been carefully revised by the authors. As far as I can see (and as far as it was realistically possible) all my previous comments have been appropriately addressed. In consequence, this work is now of much higher quality and relevance. Importantly, this manuscript now provides strong evidence for a real-world relevance of its major findings and the resulting model (currently Supp. Fig. 9) is now well supported, novel and of general interest.

Nevertheless, there is room for improvement. In this regard, I have the following comments (not sorted by importance):

- l67: "acts as a key trigger" non-supported overstatement, delete "key", consider reformulation to avoid "trigger"

- introduction: not very balanced, very detailed on VOCs, rudimentary on calcium. This work is for a general audience and the authors cannot expect that every reader would be familiar with details and principles of plant calcium signaling. Around line 68 they should at least add one explanatory sentence on calcium signal implementation/decoding and should quote one or two recent reviews, which a reader could consult.

- In my opinion the Figure headlines should provide a clear and concise statement about the major finding/conclusion of the data instead of a rather technical description. This refers especially to Figs.4, 5, 6 and 7.
- Figures in general: It is now time to consider their appearance in "print". Currently, the format, assembly and also to some extent the proportion of their elements is rather imperfect. Most figures are at intermediate scale between single column and double column stile. This would be a nightmare for any careful typesetter. Single column presentation would be too small for most figures to allow recognizing details, double column presentation would waste space. All Figures require extensive improvements.
- Fig.1. The authors may consider to divide 1a-c and 1d-g into two figures. Otherwise, this figure must be rearranged so that two separate modules become appreciable.
- Fig2 (and others) a mix of vertical and horizontal labels. (Z-3-HAL labeling is wasting a lot of space. I would transfer the labeling into the picture and use the space to enlarge the pictures)
- Fig 6 reorganize to avoid large white areas
- I would recommend to present the model (supp. Fig.9) as last main figure.
- Discussion: The discussion is now longer than the results part. This imbalance needs to be adjusted. Moreover, especially the first part is largely repetitive with the results, discusses existing knowledge like a review article and does not show a clear logical flow (l.196-263). The paragraphs l264.272 and l273-287 are much too detailed for a discussion in a general journal. A short excerpt may go to the results and the details may be incorporated in some methods chapter. I would recommend to build a logical path that culminates in the discussion of the newly emerging model (supp. Fig.9).

Reviewer #3 (Remarks to the Author):

I appreciate the effort of the authors to include more experiments and to quantify the VOCs released by disrupted tissues. Nevertheless the work is still repeating what has been demonstrated year ago although with a different technology. In plant insect interactions an important role is played by the insect oral secretions, and a recent paper using the same calcium reporting system shows that the insect bites and not only the insect wounding released calcium signatures (<https://doi.org/10.3390/plants11202689>). In this contexts, the experiment with homogenized tissues is completely unrealistic because it is only based on the natural emission of GLVs upon interaction of lithic enzymes with the cell membranes. An experiment with insect oral secretion would be more realistic.

I remain in my first decision that this work, although adding some additional proof of what is already know does not represent a novelty in the general concept of VOC-dependent calcium signalling.

As requested in my first revision, only a full genome analysis can provide evidence that VOCs trigger the responses expected by a calcium signal transmission pathway. The data provided are not enough for a publication on Nature Communication.

Reply to Reviewers.

We greatly appreciate the Editor and Reviewers for their encouraging comments and thoughtful reviews, which helped us to significantly improve our manuscript. We have addressed all of the raised points, and the specific responses are provided in a point-by-point manner.

Reviewer #1 (Remarks to the Author):

This new version of the manuscript is much more accurate in the presentation of data and importantly includes new experiments and analyses that have improved its quality.

The story is now more solid.

Having said that, I have identified a few minor things that require authors' attention.

Line 183. "stomatal" instead of "stomata".

Line 183. "...as well as stomatal mutants exhibiting a constitutive stomata opening phenotype...". This is not really correct for SLAC1. It was shown that the slac1 mutant is also impaired in the stomatal opening as reported in Laanemets et al. (New Phytologist (2013) 197: 88–98 doi: 10.1111/nph.12008).

We appreciate your suggestion. We have revised the text accordingly.

Revised main text: Page 6: To examine the function of stomata in rapid GLV sensory transduction, we used the phytohormone abscisic acid (ABA) to induce stomatal closure⁴⁰, as well as stomatal mutants exhibiting abnormal stomatal movement phenotypes^{41, 42, 43}.

Line 258. "These results suggest that in GLV responses, there is a possibility that GLVs do not activate the key elements required for long-distance signal transmission, such as GLRs, or that signals cannot reach the core in the vasculature".

Based on our direct experience, to induce long-distance signals in response to wounding you need to damage the main vein. The distance the SWP can cover depends on the magnitude of the stimulus. It is thus plausible that VOCs do not generate a local response sufficient to reach the threshold required to allow the SWP to move out of the local leaf.

We thank you for your suggestion. We have mentioned this possibility in the Discussion section of the revised manuscript.

Revised main text: Page 8: Given that the amplitude and propagation rate of the wound-triggered systemic signals depend on the type and intensity of stimuli⁶¹ and the necessity of damaging the main vein of the leaf⁶², GLVs might not activate the key elements required for long-distance signal transmission, such as GLRs, or the local response might not reach the threshold required to allow the Ca²⁺/electrical signals to move out of the local leaf.

Reviewer #2 (Remarks to the Author):

The newly submitted manuscript has been carefully revised by the authors. As far as I can see (and as far as it was realistically possible) all my previous comments have been appropriately addressed. In consequence, this work is now of much higher quality and relevance. Importantly, this manuscript now provides strong evidence for a real-world relevance of its major findings and the resulting model (currently Supp. Fig. 9) is now well supported, novel and of general interest.

Nevertheless, there is room for improvement. In this regard, I have the following comments (not sorted by importance):

- l67: “acts as a key trigger” non-supported overstatement, delete “key”, consider reformulation to avoid “trigger”

We are grateful for these comments. We have revised the text accordingly.

Revised main text: Page 3: These findings suggest that Ca²⁺ signaling is involved in an early process leading to downstream defense responses.

- introduction: not very balanced, very detailed on VOCs, rudimentary on calcium. This work is for a general audience and the authors cannot expect that every reader would be familiar with details and principles of plant calcium signaling. Around line 68 they should at least add one explanatory sentence on calcium signal implementation/decoding and should quote one or two recent reviews, which a reader could consult.

We appreciate your valuable comment. We have provided an explanation concerning the role of Ca²⁺ signaling in plant stress responses and appropriately referenced relevant research in the Introduction section.

Revised main text: Page 2: Cytosolic Ca²⁺ plays crucial roles in a wide array of plant stress responses^{26, 27}. Downstream signaling activation systems via cytosolic calcium ion concentration ([Ca²⁺]_{cyt}) increases have been extensively studied, including the identification of critical components linking stress perception to the formation of Ca²⁺ signals, as well as the mechanism of Ca²⁺ signal implementation²⁸.

- In my opinion the Figure headlines should provide a clear and concise statement about the major finding/conclusion of the data instead of a rather technical description. This refers especially to Figs.4, 5, 6 and 7.

We are grateful for your suggestion. We have updated the titles of Figs. 4–7, which are now designated as Figs. 5–8.

Revised main text: Page 19: Fig. 5. Z-3-HAL-induced Ca²⁺ signature occurs

locally and in a concentration-dependent manner

Revised main text: Page 19: $[Ca^{2+}]_{cyt}$ increases in guard, mesophyll, and vasculature cells, and subsequently in epidermal cells upon Z-3-HAL exposure

Revised main text: Page 19: Upright confocal laser scanning microscopy reveals instant Z-3-HAL-induced $[Ca^{2+}]_{cyt}$ increases in guard cells, followed by mesophyll cells

Revised main text: Page 20: Stomatal opening plays a crucial role in $[Ca^{2+}]_{cyt}$ increases upon Z-3-HAL exposure

- Figures in general: It is now time to consider their appearance in “print”. Currently, the format, assembly and also to some extent the proportion of their elements is rather imperfect. Most figures are at intermediate scale between single column and double column stile. This would be a nightmare for any careful typesetter. Singe column presentation would be too small for most figures to allow recognizing details, double column presentation would waste space. All Figures require extensive improvements.

We appreciate your suggestions. We have modified all figures in the manuscript to ensure their fit within the frames of both single-column and double-column formats.

- Fig.1. The authors may consider to divide 1a-c and 1d-g into two figures. Otherwise, this figure must be rearranged so that two separate modules become appreciable.

We thank you for this suggestion. We have divided Fig. 1a–c and Fig. 1d–g into new Figs. 1 and 2, respectively.

- Fig2 (and others) a mix of vertical and horizontal labels. (Z-3-HAL labeling is wasting a lot of space. I would transfer the labeling into the picture and use the space to enlarge the pictures)

We thank you for your comment. We have moved the labeling (e.g., Z-3-HAL) into the pictures in all figures.

- Fig 6 reorganize to avoid large white areas

We appreciate this comment. We have redesigned Fig. 6 (new Fig. 7).

- I would recommend to present the model (supp. Fig.9) as last main figure.

We are grateful for your remarks. We have moved Supp. Fig. 9 to Figure 9.

- *Discussion: The discussion is now longer than the results part. This imbalance needs to be adjusted. Moreover, especially the first part is largely repetitive with the results, discusses existing knowledge like a review article and does not show a clear logical flow (l.196-263). The paragraphs l264.272 and l273-287 are much too detailed for a discussion in a general journal. A short excerpt may go to the results and the details may be incorporated in some methods chapter. I would recommend to build a logical path that culminates in the discussion of the newly emerging model (supp. Fig.9).*

We thank you for these valuable suggestions. We have revised the Discussion accordingly. We have significantly reduced the word count from 1731 to 1341 words, particularly in the first half of the Discussion, making it shorter than Results (1675 words). Furthermore, we have revised the structure of Discussion to highlight the model (new Fig. 9) illustrating the entry of GLVs into plant tissues.

Reviewer #3 (Remarks to the Author):

I appreciate the effort of the authors to include more experiments and to quantify the VOCs released by disrupted tissues. Nevertheless the work is still repeating what has been demonstrated year ago although with a different technology. In plant insect interactions an important role is played by the insect oral secretions, and a recent paper using the same calcium reporting system shows that the insect bites and not only the insect wounding released calcium signatures (<https://doi.org/10.3390/plants11202689>). In this contexts, the experiment with homogenized tissues is completely unrealistic because it is only based on the natural emission of GLVs upon interaction of lithic enzymes with the cell membranes. An experiment with insect oral secretion would be more realistic.

I remain in my first decision that this work, although adding some additional proof of what is already know does not represent a novelty in the general concept of VOC-dependent calcium signalling.

As requested in my first revision, only a full genome analysis can provide evidence that VOCs trigger the responses expected by a calcium signal transmission pathway. The data provided are not enough for a publication on Nature Communication.

We agree with your comment that the use of insect oral secretions capable of mimicking herbivore feeding and inducing plant defense reactions including VOC emission would serve as a robust approach closely aligned with natural conditions. Hence, we redesigned our experimental approach using the herbivorous insect *S. litura*, drawing inspiration from the methodology of Dr. Maffei's team (Zebelo et al., *Plant Sci.*, 2012; Fig. 1a) to align more closely with the natural conditions. Our new experiment revealed that VOCs emitted by feeding insects, possibly including oral secretions, elicit Ca²⁺ signals in receiver *Arabidopsis* (Fig. 1b and c). Furthermore, based on the quantification of VOCs, we stated that the observed Ca²⁺ increases in our experimental settings could occur under natural conditions. Although we did not focus on insect secretions in this study, an experiment using oral secretions would be an important task for future studies on plant–insect interactions.

We sincerely appreciate your valuable comment regarding full-genome analysis. As mentioned in the Discussion section, previous studies investigated the comprehensive transcriptional changes induced by GLVs in *Arabidopsis*, demonstrating that a diverse range of signaling pathways can be activated (Yamauchi et al., *J. Pestic. Sci.*, 2018). Although we acknowledge that full-genome analysis could unravel GLV-induced transcriptional networks related to stress responses, precisely pinpointing the exact signaling pathway linked to Ca²⁺ signals via full-genome analysis is challenging at the present stage. We have mentioned the importance of full-genome analysis and emphasized its potential role in further elucidating the complex signaling pathway activated by Ca²⁺ signals.

Revised main text: Page 7: Different experimental approaches, such as transcriptome analysis, could provide new insights into the mechanism underlying enhanced stress tolerance through GLV-induced Ca²⁺ signals.